# Mutation of *vsx* genes in zebrafish highlights the robustness of the retinal specification network

Joaquín Letelier[1,2]*[†], Lorena Buono[1,3][†], María Almuedo-Castillo[1], Jingjing Zang[4], Constanza Mounieres[2], Sergio González-Díaz[1], Rocío Polvillo[1], Estefanía Sanabria-Reinoso[1], Jorge Corbacho[1], Ana Sousa-Ortega[1], Ruth Diez del Corral[5], Stephan CF Neuhauss[4], Juan R Martínez-Morales[1]*

[1]Centro Andaluz de Biología del Desarrollo (CSIC/UPO/JA), Sevilla, Spain; [2]Centre for Integrative Biology, Facultad de Ciencias, Universidad Mayor, Santiago, Chile; [3]IRCCS SYNLAB SDN, Via E. Gianturco, Naples, Italy; [4]Department of Molecular Life Sciences, University of Zürich, Zürich, Switzerland; [5]Champalimaud Research, Champalimaud Centre for the Unknown, Lisbon, Portugal

**\*For correspondence:**
joaquin.letelier@umayor.cl (JL);
jrmarmor@upo.es (JRM-M)

[†]These authors contributed equally to this work

**Competing interest:** The authors declare that no competing interests exist.

**Abstract** Genetic studies in human and mice have established a dual role for *Vsx* genes in retina development: an early function in progenitors' specification, and a later requirement for bipolar-cells fate determination. Despite their conserved expression patterns, it is currently unclear to which extent *Vsx* functions are also conserved across vertebrates, as mutant models are available only in mammals. To gain insight into *vsx* function in teleosts, we have generated *vsx1* and *vsx2* CRISPR/Cas9 double knockouts (*vsx*KO) in zebrafish. Our electrophysiological and histological analyses indicate severe visual impairment and bipolar cells depletion in *vsx*KO larvae, with retinal precursors being rerouted toward photoreceptor or Müller glia fates. Surprisingly, neural retina is properly specified and maintained in mutant embryos, which do not display microphthalmia. We show that although important *cis*-regulatory remodelling occurs in *vsx*KO retinas during early specification, this has little impact at a transcriptomic level. Our observations point to genetic redundancy as an important mechanism sustaining the integrity of the retinal specification network, and to *Vsx* genes regulatory weight varying substantially among vertebrate species.

## Editor's evaluation

This study provides important insights into how tissue specification networks, while often employing conserved genes across species, can differ in their network architecture, resulting in differences in how they buffer perturbations. This is shown for the Visual System Homeobox genes (VSX) in the zebrafish retinal specification pathway, where yet-to-be-defined compensatory mechanisms prevent microphthalmia in the absence of VSX function, something not observed in humans or mice. The evidence supporting the conclusions of the study is solid and provides a foundation for further molecular and genetic analysis of retinal specification. This work is relevant to developmental biologists with interests in tissue specification and gene regulatory networks.

## Introduction

The organogenesis of the vertebrate eye is a complex multistep process entailing the sequential activation of genetic programs responsible for the initial specification of the eye field, the patterning of the eye primordium into sub-domains, and the determination of the different neuronal types.

Although we are far from understanding the precise architecture of the gene regulatory networks (GRNs) controlling eye formation, many of their central nodes have been already identified (*Buono and Martinez-Morales, 2020*; *Fuhrmann, 2010*; *Heavner and Pevny, 2012*; *Martinez-Morales, 2016*). They comprise transcriptional regulators recruited repeatedly for key developmental decisions at different stages of eye formation, and which mutation in humans is often associated to severe ocular malformations: that is, microphthalmia, anophthalmia, and coloboma. This is the case for SIX3, PAX6, RAX, SOX2, VSX2, or OTX2 (*Gregory-Evans et al., 2004*; *Gregory-Evans et al., 2013*).

Among the main regulators, the visual system homeobox transcription factors, Vsx1 and Vsx2, have been shown to control the development of visual circuits in vertebrate and invertebrate species (*Burmeister et al., 1996*; *Erclik et al., 2008*; *Focareta et al., 2014*). *Vsx2*, initially termed as *Chx10*, was the first gene of the family characterized in vertebrates (*Liu et al., 1994*). *Vsx2/Chx10* shows a conserved expression pattern across vertebrate species, both in the retina (i.e. early in all optic cup precursors, and later in retinal bipolar cells), as well as in hindbrain and spinal cord interneurons (*Ferda Percin et al., 2000*; *Kimura et al., 2013*; *Liu et al., 1994*; *Passini et al., 1997*). A nonsense mutation in *Vsx2* (Y176stop) turned to be the molecular cause of the phenotype exhibited by the classical mutant mice *ocular retardation* (*or*), which displays microphthalmia and optic nerve aplasia (*Burmeister et al., 1996*; *Truslove, 1962*). The phenotypic analysis of *or* mutants, as well as the examination of human patients with hereditary microphthalmia, revealed an essential role for *Vsx2* in neuro-epithelial proliferation and bipolar cells differentiation (*BarYosef et al., 2004*; *Burmeister et al., 1996*; *Ferda Percin et al., 2000*). Subsequent studies indicated that, during optic cup formation, *Vsx2* is a key factor in the binary decision between neural retina and retinal-pigmented epithelium (RPE) lineages. Genetic studies in mice and chick revealed that *Vsx2* acts, downstream of the neural retina inducing ligands (i.e. FGFs), as a repressor of *Mitf* and *Tfec* genes and hence of the RPE identity (*Horsford et al., 2005*; *Nguyen and Arnheiter, 2000*; *Rowan et al., 2004*).

A few years after *Vsx2* identification, a closely related paralog, *Vsx1*, was reported in several vertebrate species (*Chen and Cepko, 2000*; *Chow et al., 2001*; *Levine et al., 1994*; *Passini et al., 1997*). The proteins encoded by these paralogous genes have similar domains' architecture, including well-conserved paired-like homeodomain and CVC (Chx10/Vsx-1 and ceh-10) regulatory modules, and share biochemical properties, binding with high affinity to the same DNA sequence motif 'TAATTAGC' (*Capowski et al., 2016*; *Dorval et al., 2005*; *Ferda Percin et al., 2000*; *Heon, 2002*). Although both genes display partially overlapping expression patterns in the retina, *Vsx2* precedes *Vsx1* expression in undifferentiated progenitors in all vertebrate models analysed. Furthermore, once retinal precursors exit the cell cycle, they are expressed in complementary sets of differentiated bipolar cells. Thus, *Vsx1* is restricted to different types of ON and OFF cone bipolar cells in mice, and *Vsx2* to S4 bipolar and Müller cells in zebrafish (*Ohtoshi et al., 2004*; *Shi et al., 2011*; *Vitorino et al., 2009*). In contrast to *Vsx2*, *Vsx1* seems to have a minor contribution to retinal specification in mammals. A single case of sporadic microphthalmia has been associated to *Vsx1* mutation in humans (*Matías-Pérez et al., 2018*), and its mutation in mice does not affect early retinal development even in a *Vsx2* mutant background (*Chow et al., 2004*; *Clark et al., 2008*). However, *Vsx1* mutation has been linked to inherited corneal dystrophies in humans, and is associated to abnormal electroretinogram (ERGs) recordings either in mice or in patients (*Chow et al., 2004*; *Heon, 2002*; *Mintz-Hittner et al., 2004*).

Despite all these advances on the developmental role of *Vsx* genes, many questions remain open. A fundamental issue is to understand to which extent *Vsx* gene functions are conserved across vertebrates. Previous antisense oligonucleotides or morpholino studies in zebrafish have shown that *vsx2* knockdown results in microphthalmia and optic cup folding defects (*Barabino et al., 1997*; *Clark et al., 2008*; *Gago-Rodrigues et al., 2015*; *Vitorino et al., 2009*). However, these findings have not been validated using knockout lines, neither the role of *vsx1* and *vsx2* in fate determination and bipolar cells differentiation has been sufficiently explored in teleost fish.

To gain insight into the universality and diversity of *Vsx* functions, we have generated zebrafish mutants for *vsx1* and *vsx2* harboring deletions within the homeodomain-encoding exons. Surprisingly, eye morphology and size appear normal either in the individual or in the double *vsx1/vsx2* mutants, thus indicating that *vsx* genes are not essential to initiate retinal development in zebrafish. The absence of early retinal malformations facilitates the phenotypic analysis of the mutants at later embryonic and larval stages. Defects in the visual background adaptation (VBA) reflex are observed in *vsx1* mutant, and appear enhanced in double mutant larvae, suggesting partial or complete blindness.

Analysis of ERG responses confirms vision loss, showing that the amplitude of the b-wave recordings is reduced in *vsx1* mutants, and absent in double mutants. Interestingly, a single wild type copy of *vsx1* is sufficient to prevent VBA and ERG defects, indicating that *vsx2* loss of function can be compensated by *vsx1*. The analysis of neuronal-specific markers confirmed that retinal progenitors fail to differentiate into bipolar cells in double mutant embryos. Instead, we show that precursors at the inner nuclear layer (INL) can remain proliferative, undergo apoptosis, or be rerouted toward other retinal lineages, particularly differentiating as Müller glial cells. Finally, we investigate whether transcriptional adaptation (*El-Brolosy et al., 2019*) may compensate for *vsx1/vsx2* loss-of-function during retinal specification. The transcriptomic analysis of core components of the retinal specification GRN do not support a transcriptional adaptation mechanism in *vsx1/vsx2* double mutants, rather suggesting that the network robustness is by itself sufficient to sustain early eye development even in the absence of *vsx1* and *vsx2* function. In summary, whereas our work shows a conserved role for *Vsx* genes during bipolar cell differentiation, also indicates that their hierarchic weight within the eye GRNs varies considerably across vertebrate species.

## Results

### Zebrafish *vsx* double mutants show normal eye size but affected lamination of the retina

Despite the additional round of genome duplication occurring in the teleost lineage after the split with sarcopterygians (*Meyer and Schartl, 1999*), a single copy of both *vsx1* and *vsx2* was retained in zebrafish. In order to investigate the role of Vsx transcription factors during visual system formation in zebrafish, we generated mutants for both paralogs using CRISPR/Cas9. To optimize the generation of null animals, we targeted conserved regions encoding for the DNA binding domain of the proteins in their corresponding loci at chromosome 17 (*Figure 1a*). We generated a 245 bp deletion in *vsx1* encompassing exon3, intron3, and exon4 of the gene (*vsx1Δ245*). This mutation results in an in-frame deletion of 53 amino acids by the removal of 159 bp from exon3 (54 bp) and exon4 (105 bp; *Figure 1—figure supplement 1a*). In the case of *vsx2*, a 73 bp deletion was generated in exon 3 (*vsx2Δ73*). This mutation deletes 24 amino acids of the core DBD of the protein and generates a premature stop codon in that domain (*Figure 1—figure supplement 1b*). Both deletions can be easily screened by PCR with primers flanking the mutation sites. Using Vsx1- and Vsx2-specific antibodies, we found that no Vsx2 or Vsx1 proteins could be detected by western blot in 24hpf *vsx*KO samples (*Figure 1—figure supplement 1c, d*). In addition, no maternal Vsx1 protein was detected in early 1.5hpf wildtype embryos (*Figure 1—figure supplement 1c*).

At 2-week post fertilization, no obvious macroscopic defects were observed in the visual system of either homozygous single mutants (i.e. *vsx1Δ245* or *vsx2Δ73*) or homozygous double mutants *vsx1Δ245; vsx2Δ73* (here termed *vsx*KO), which appeared normal in shape and size (*Figure 1—figure supplement 2*; *Figure 1—figure supplement 3a–d*). Homozygous single mutants, and even animals harboring a single wild type copy either of *vsx1* (*vsx1Δ245+/-, vsx2Δ73-/-*) or *vsx2* (*vsx1Δ245-/-; vsx2Δ73+/-*) reached adulthood and were fertile. However, double mutant larvae (*vsx1Δ245 -/-; vsx2Δ73 -/-*) died at around 3-week post fertilization, with the exception of a single unfertile escaper reaching adulthood (1 out of 152 larvae raised). For further analyses, double mutant embryos and larvae were obtained each generation by in-crossing of *vsx1Δ245+/-; vsx2Δ73-/-* or *vsx1Δ245-/-; vsx2Δ73+/-*-animals. Once the proper recombinants were obtained, heterozygous lines maintenance was facilitated by the linkage between *vsx1* and *vsx2* mutant alleles, which tend to segregate together due to their proximity (10.6 Mb) in chromosome 17.

Histological sectioning of mutant retinas at 48hpf showed a small delay in the formation of the inner plexiform layer (IPL), but no obvious macroscopic optic cup malformations when compared to WT (*Figure 1b and f*). At 72hpf, both the outer plexiform layer (OPL) and the IPL appeared less organized in the double mutant retinas, which showed discontinuities/fenestrae (*Figure 1c and g*). At 6dpf, double mutant larvae showed all the layers of a normal retina, but the thickness of the outer (ONL) and inner (INL) nuclear layers was significantly increased and reduced respectively, when compared to siblings (*Figure 1d and h*; *Figure 1—figure supplement 3e, h, i*). In addition to retinal layer formation defects, *vsx*KO fish presented expanded pigmentation in skin melanocytes even when exposed to bright light for 20 min (*Figure 1e and i*; *Figure 1—figure supplement 3a, d*). This phenomenon

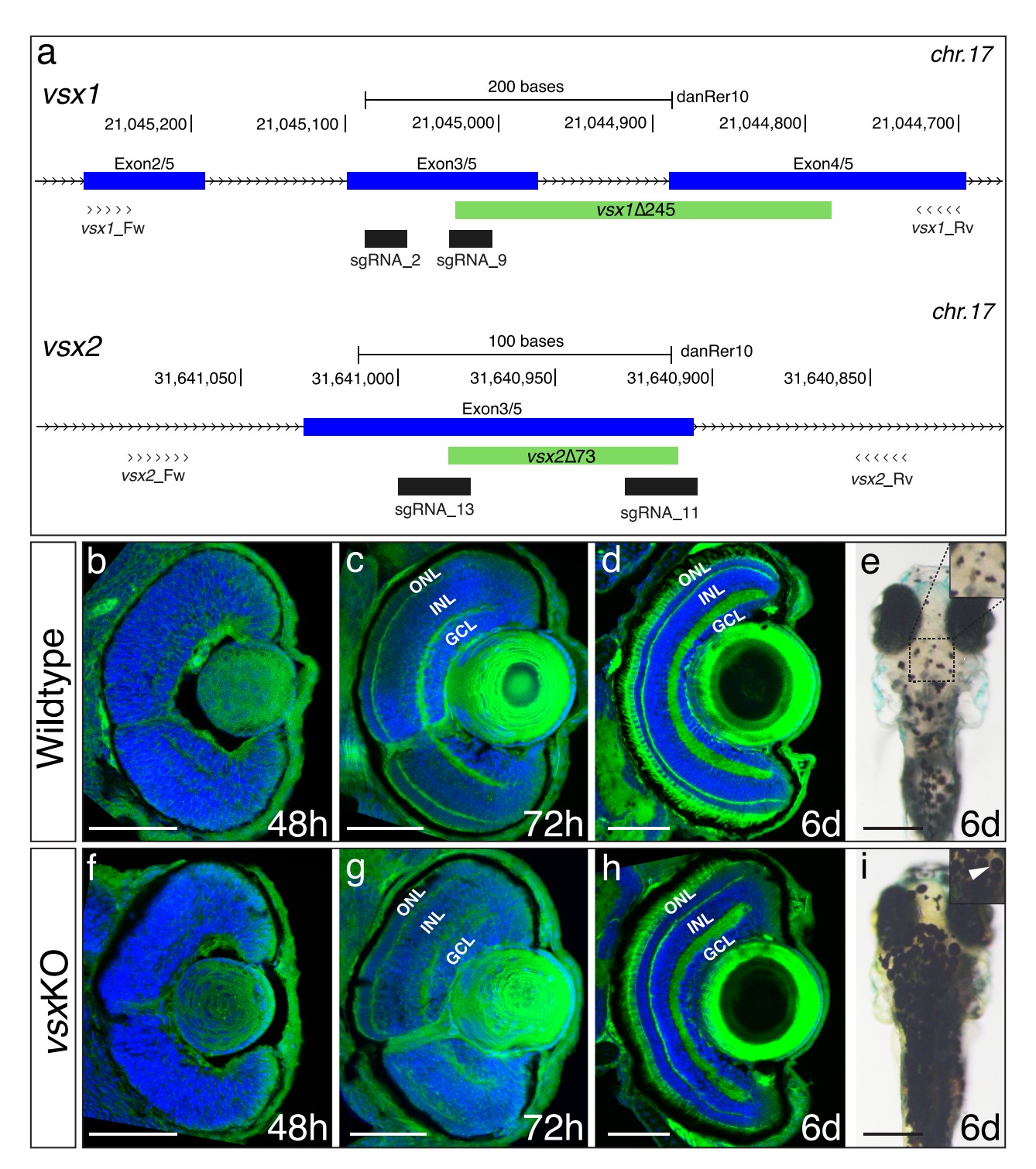

**Figure 1.** DNA-binding domain deletion of *vsx* genes affect neural retina formation and disrupt VBA reflex. (**a**) CRISPR/Cas9 DNA editing tool was used to generate deletions (green box) in the highly conserved DBD from *vsx1* (top) and *vsx2* (bottom) TFs. Blue boxes represent gene exons, black boxes the location of sgRNAs used to guide Cas9 endonuclease and primers for screening are depicted as opposing arrowheads. **b-d** and **f-h**. Histological sections stained with nuclear marker DAPI and phalloidin-Alexa488 for actin filaments from WT (**b-d**, n≥8) and *vsx*KO central retinas (**f-h**, n≥10) at 48hpf (**b, f**), 72hpf (**c, g**) and 6dpf (**d, h**). (**e, i**). Head dorsal view from 6dpf WT (**e**) and *vsx*KO (**i**) larvae with insets showing their pigmentation pattern (white

*Figure 1 continued on next page*

*Figure 1 continued*

arrowhead). ONL: outer nuclear layer, INL: inner nuclear layer, GCL: ganglion cell layer, hpf: hours post-fertilization, dpf: days post-fertilization. Scale bar in (**b-d**) and (**f-h**): 50 μm, scale bar in **e** and **i**: 500 μm.

The online version of this article includes the following source data and figure supplement(s) for figure 1:

**Figure supplement 1.** Zebrafish Vsx1 and Vsx2 proteins are disrupted in *vsx*KO animals.

**Figure supplement 1—Source data 1.** Raw unedited western blot gel for Vsx1.

**Figure supplement 1—Source data 2.** Raw unedited western blot gel for Vsx2.

**Figure supplement 1—Source data 3.** Uncropped Vsx1 blot with labelled bands.

**Figure supplement 1—Source data 4.** Uncropped Vsx2 blot with labelled bands.

**Figure supplement 2.** Eye size is normal in *vsx*KO juvenile fish.

**Figure supplement 3.** VBA and nuclear layers width are affected in *vsx* mutants.

is indicative of an impaired visual background adaptation (VBA) reflex, and is often associated with blindness in zebrafish (*Fleisch and Neuhauss, 2006*).

## Visual function is impaired in single *vsx1* and *vsx*KO double mutants

To test the visual performance of the *vsx* mutants; ERG recordings were obtained from WT and mutants at 5 dpf (*Figure 2*). Zebrafish retina becomes fully functional at 5 dpf with the exception of late maturing rods (*Bilotta et al., 2001*) and thus, the recorded field potentials were mainly contributed by cones. Wild type larvae show a standard ERG response to light flash, characterized by a large positive b-wave representing the depolarization of ON bipolar cells (*Figure 2a*), which also masks the initial a-wave generated by photoreceptor (PR) hyperpolarization. Representative recordings

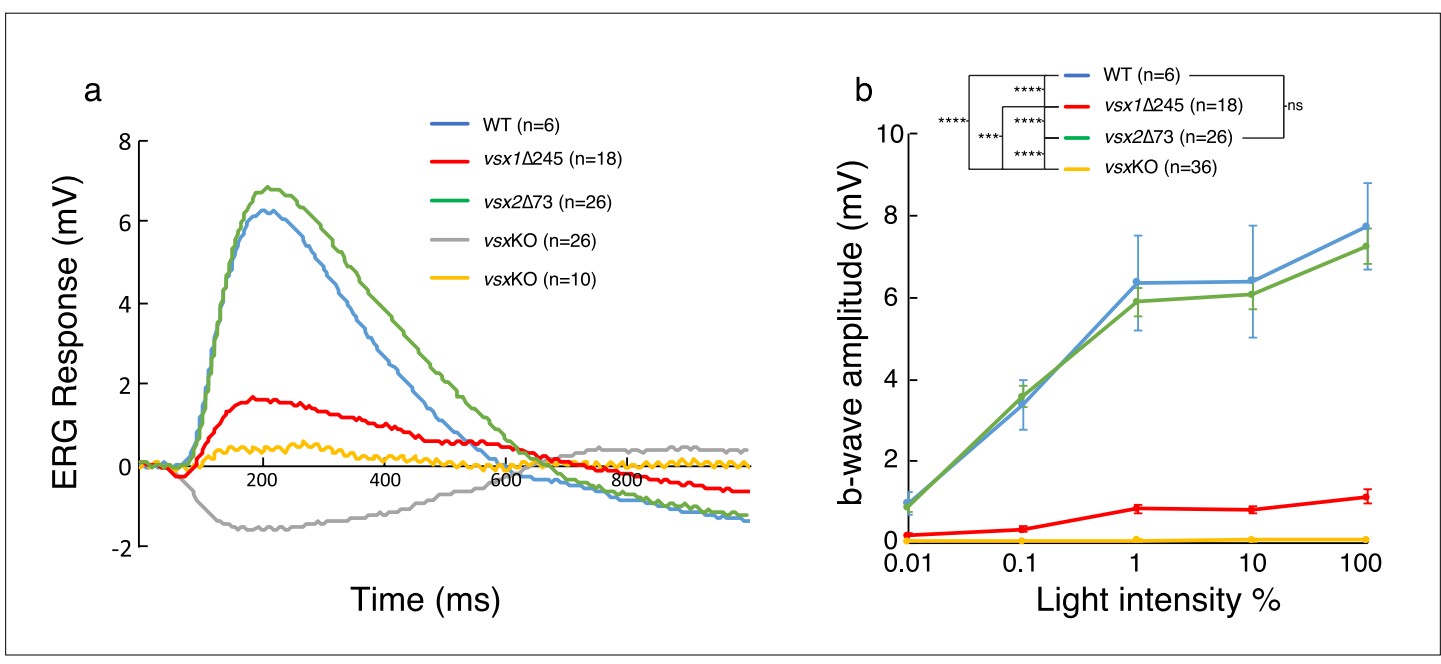

**Figure 2.** ERG response is reduced in *vsx*KO larvae. (**a**) Representative ERG tracks at maximum light intensity from WT (blue), *vsx1*Δ245 (red), *vsx2*Δ73 (green) and *vsx*KO double mutants (grey and yellow) at 5dpf. For *vsx*KO larvae, two typical recordings are shown (grey and yellow tracks). (**b**). Averaged ERG b-wave amplitudes from WT (blue), *vsx1*Δ245 (red), *vsx2*Δ73 (green) and *vsx*KO (yellow) larvae. No significant differences were observed between WT and *vsx2*Δ73 samples. *vsx1*Δ245 and *vsx*KO mutants produce a significant reduction of the ERG b-wave amplitude compared with both WT and *vsx2*Δ73 larvae throughout all light intensities tested (***p<0.0001, ****p<0.00001). Data are shown as mean ± SEM. In (**a**) and (**b**), *vsx1*Δ245 (red tracks) represents both *vsx1*Δ245-/- and *vsx1*Δ245-/-; *vsx2*Δ73+/-genotypes; while *vsx2*Δ73 (green tracks) represents both *vsx2*Δ73-/- and *vsx1*Δ245+/-; *vsx2*Δ73-/- genotypes. Data were collected from five independent experiments. For statistical comparison, one way ANOVA test was used. ms: milliseconds, mV: millivolts.

The online version of this article includes the following figure supplement(s) for figure 2:

**Figure supplement 1.** OKR measurements indicate decreased eye movement velocity in *vsx* mutants.

from larvae harboring different *vsx* genotypes are shown in *Figure 2a*. We found that *vsx2Δ73* ERG response (green curve) was similar to the WT recording (blue curve). However, recordings in *vsx1Δ245* larvae showed a reduced b-wave compared to WT or *vsx2Δ73* larvae. From the 36 double mutant larvae recorded in total, 10 of them still showed a b-wave, though reduced in comparison to *vsx1Δ245* mutants, and much smaller than WT recordings. Moreover, in the remaining 26 double mutants recorded only the negative a-wave but not the b-wave (gray curve) was detected, suggesting that bipolar cells differentiation and/or function might be compromised. Statistical analysis of the average amplitude showed that the b-wave is significantly decreased in both *vsx1Δ245* single and *vsx*KO double mutants in comparison to WT at all tested light intensities (*Figure 2b*). In addition, the b-wave response amplitude in the double mutant was significantly reduced compared to *vsx1* single mutants (*Figure 2b*). These measurements are in line with our previous observation indicating that double mutant retinas are more affected at the cellular level than single *vsx1Δ245* animals (*Figure 1—figure supplement 3*).

To quantitatively characterize eye performance, optokinetic response (OKR) recordings (*Rinner et al., 2005*) were obtained for WT and *vsx* mutant fish (*Figure 2—figure supplement 1*). To investigate the role of Vsx transcription factors at the behavioral level, eye movement velocity was recorded at 5dpf in WT and *vsx*KO mutant fish. We measured eye velocity varying different parameters of the moving stimuli, such as contrast (contrast sensitivity; *Figure 2—figure supplement 1a*), frequency (spatial resolution; *Figure 2—figure supplement 1b*) and angular velocity (temporal resolution; *Figure 2—figure supplement 1c*). In all conditions tested, we observed a significant reduction in eye velocity for *vsx1* single and *vsx*KO double mutants when compared with *vsx2Δ73* larvae and WT controls (repeated measurement, ANOVA p<0.001). Taken together these physiological recordings confirmed significant sight impairment in *vsx1* mutants, a phenotype that is further aggravated by *vsx2* loss in *vsx*KO double mutants.

## Extended proliferation wave and INL cell death in *vsx*KO double mutant retinas

As *vsx*KO double mutants showed stronger retinal architecture and visual defects than other *vsx* mutant combinations, we decided to focus further phenotypic analyses on them. To assess whether our observations on the increased thickness of the ONL and the decreased width of the INL (*Figure 1—figure supplement 3*) correlate with a proliferation and/or cell death unbalance, we examined both parameters in *vsx*KO fish. To investigate proliferation defects, we quantified the number of phospho-histone H3 positive (PH3+) cells in the retina of wild type and *vsx*KO animals throughout the lamination process: that is, at 24, 48, 60, and 72hpf (*Figure 3a–f and m*; *Figure 3—figure supplement 1*). At 24 and 48hpf, no difference in the number of PH3 + cells were observed between WT and *vsx*KO retinas (*Figure 3a, d and m*; *Figure 3—figure supplement 1*). However, at 60hpf, when the proliferation wave has largely finished in WT eyes, double mutant retinas continued to divide and showed a significant increase in PH3 + cells, particularly in the outer and peripheral regions (*Figure 3b, e and m*). Later on, at 72hpf, PH3 + cells were only detected in the CMZ and no significant difference in the number of proliferative cells was detected between WT and *vsx*KO retinas (*Figure 3c, f and m*). To test if cell death may account for the reduced INL width observed in double mutants (*Figure 1—figure supplement 3h, i*), we stained retinal cryosections at different time points with anti-cleaved caspase3 (C3) antibodies to detect cells that undergo apoptosis (*Figure 3g–l and n*). At 60hpf, C3-positive cells (C3+) could be observed rarely in WT or *vsx*KO retinas (*Figure 3g, j and n*). However, at both 72 and 96hpf, a significant increase in the number of apoptotic C3 + cells were detected in double mutants compared to WT (*Figure 3h, i, k and l*). Apoptotic cells concentrated mainly in the INL layer of the retina (*Figure 3k, l and n*), suggesting that cell death within this layer may contribute to the decreased thickness observed in *vsx*KO retinas. We also observed a few apoptotic C3 + cells in the ganglion cell layer (GCL) in *vsx*KO embryos (*Figure 3k and l*) suggesting than the survival of these cells may be compromised. To investigate this point, we decided to analyze the integrity of the retinal ganglion cells' (RGCs) projections to the optic tectum by injecting DiI and DiO tracers in WT and *vsx*KO double mutant eyes at 6dpf (*Video 1*). No obvious differences in retino-tectal projections were detected between WT and double mutant larvae, indicating that the RGCs are not affected in *vsx*KO retinas compared to control animals.

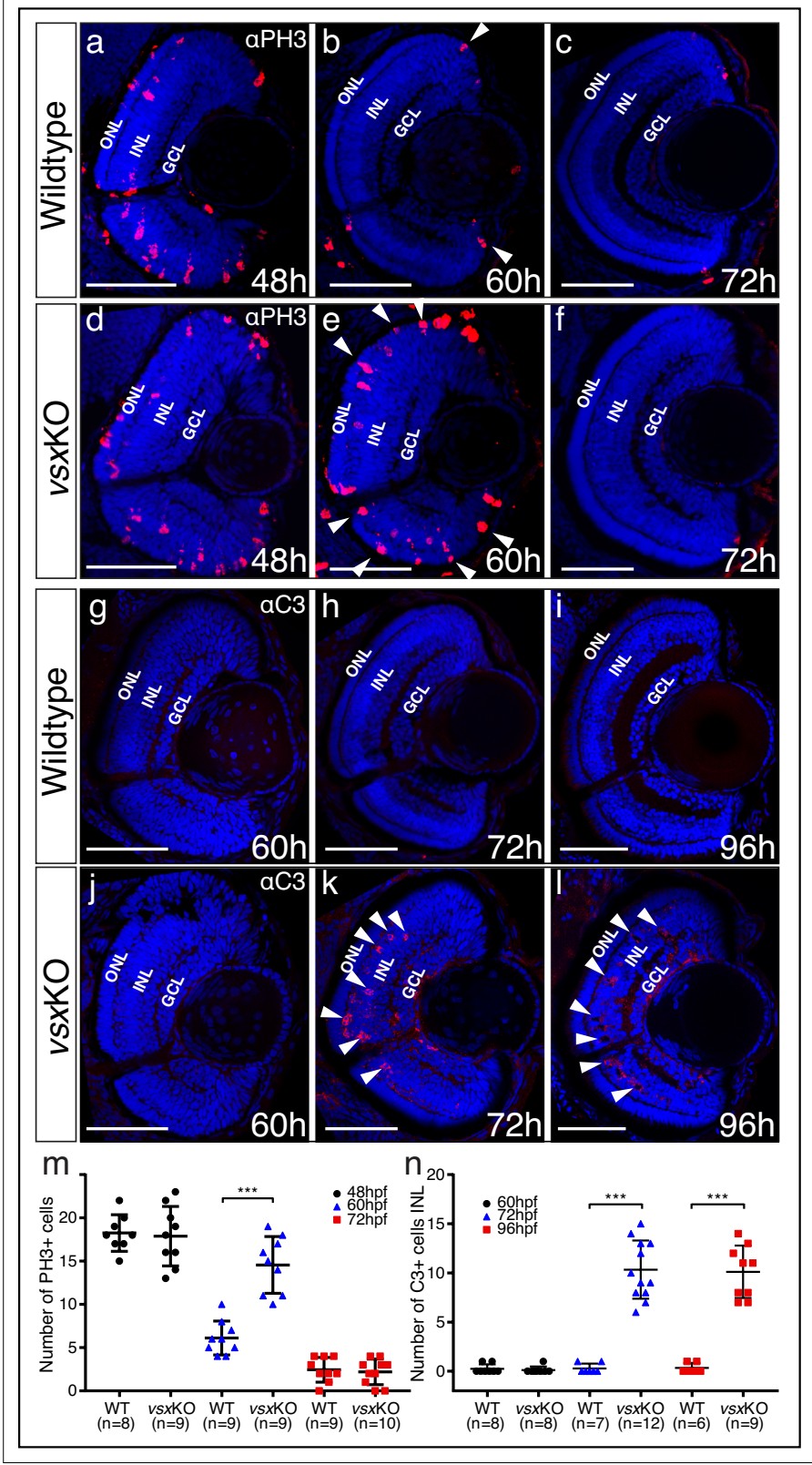

**Figure 3.** Mitosis and apoptosis markers expression are increased in *vsx*KO retinas. (**a-f**). Phospho-histone H3 (PH3) antibody staining reveals cell divisions in central retina cryosections from WT (**a-c**) and *vsx*KO (**d-f**) samples at three different developmental stages (48, 60, and 72hpf). Increased PH3 staining was observed in *vsx*KO retinas at 60hpf (white arrowheads in **e**) compared to WT samples (white arrowheads in **b**). (**g-l**). Caspase-3 (C3) antibody

*Figure 3 continued on next page*

*Figure 3 continued*

staining was used to evaluate cell death in central retina cryosections from WT (**g-i**) and *vsx*KO (**j-l**) samples at three different developmental stages (60, 72, and 96hpf). Aberrant C3 staining was observed in *vsx*KO retinas at 72 and 96hpf (white arrowheads in **k** and **l**) compared to WT samples (**h** and **i**). **m**. Quantification of PH3 positive cells in WT and *vsx*KO retinas at different stages. Using an unpaired *t*-test, a significant increase in PH3 positive cells was observed in *vsx*KO samples at 60hpf compared to WT (***p<0.0001) but no significant changes were observed at other stages analysed (48 and 72hpf). **n**. Quantification of C3 positive cells in WT and *vsx*KO retinas at different stages. Significant increase in C3-positive cells was observed in *vsx*KO samples at 72 and 96hpf compared to WT (***p<0.0001), but no change was observed at 60hpf using an unpaired *t*-test. Data is shown as mean ± SD. ONL: outer nuclear layer, INL: inner nuclear layer, GCL: ganglion cell layer, hpf: hours post-fertilization. Scale bar in (**a-l**): 50 μm.

The online version of this article includes the following figure supplement(s) for figure 3:

**Figure supplement 1.** Delayed differentiation but normal RPE and proliferation in zebrafish *vsx*KO eyes at 24-26hpf.

## Abnormal cell fate specification in the retina in *vsx*KO

Our results indicated that, in contrast to *Vsx2* early requirement in the mouse (*Burmeister et al., 1996*), *vsx* genes are not essential for the early specification of the neural retina in zebrafish (*Figure 1*; *Figure 1—figure supplement 2*). This fact facilitated the analysis of cell fate choices in *vsx*KO embryos. Although all retinal layers are present in double mutant animals (*Figure 1—figure supplement 3*), the identity of the cells within these layers required further investigation. To examine cell fate acquisition in the INL and ONL of mutant retinas, fluorescent antisense probes or antibodies for specific markers of PRs (*prdm1a*), bipolar (*prox1*, *prkcbb*), amacrine (*ptf1a*, *pax6*), and Müller glia cells (*gfap*) were examined at 48-72hpf (*Figure 4*; *Figure 4—figure supplement 2*).

### ONL/photoreceptors specification

Prdm1a (or Blimp1) is a transcription factor that has been shown to play an early role in the specification of PR identity, mainly by the suppression of bipolar cell fate genes, including *vsx2* (*Brzezinski et al., 2010*; *Katoh et al., 2010*). Conversely, *vsx2* acute knockdown by electroporation in the post-natal mouse retina triggers a bipolar to rod fate shift (*Goodson et al., 2020*; *Livne-Bar et al., 2006*). In this study, the comparative analysis of the transient marker *prdm1a* (*Wilm and Solnica-Krezel, 2005*) between wild type and *vsx*KO embryos revealed a mild downregulation in the mutants at 72 hpf (n=6) (*Figure 4a and e*), which is in agreement with the delayed differentiation of the photoreceptors we observed in *vsx*KO animals (*Figure 4—figure supplement 1*). However, when we examined terminal differentiation markers for cones (Ab Zpr-1) and rods (Ab Zpr-3) at 72 and 96hpf, a delayed differentiation of both cell types was observed in double mutant embryos (*Figure 4—figure supplement 1*). Whereas Zpr-1 and Zpr-3 staining could be detected in the entire ONL in wild type fish from 72hpf on (*Figure 4—figure supplement 1a–c, h-j*), in 72 hpf *vsx*KO embryos the staining was restricted to a few cells in the ventral retina (*Figure 4—figure supplement 1d, k*) and was only extended to the entire ONL at 96 hpf (*Figure 4—figure supplement 1e, l*). At 6dpf, there is a significant increase of Zpr1 fluorescent intensity in *vsx*KO compared to WT retinas (*Figure 4—figure supplement 1c, f, g*), but no major differences were observed in rod stain intensity (*Figure 4—figure supplement 1j, m, n*). This result suggests that PRs' differentiation

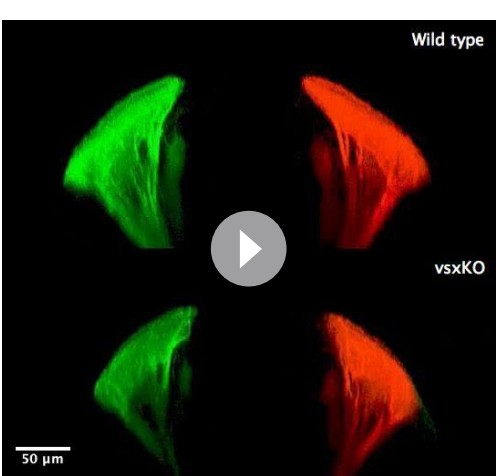

**Video 1.** *vsx*KO larvae show normal GCL retinotectal projections. (a, b). 3-D reconstructions of confocal stacks from zebrafish larval eyes injected with either DiO (green) or DiI (red) to label retinal ganglion cells and their projections to the optic tectum in wildtype (a, n=6) and *vsx*KO (b, n=8) at 6dpf. Note that *vsx*KO larvae show apparently normal retinotectal projections.
https://elifesciences.org/articles/85594/figures#video1

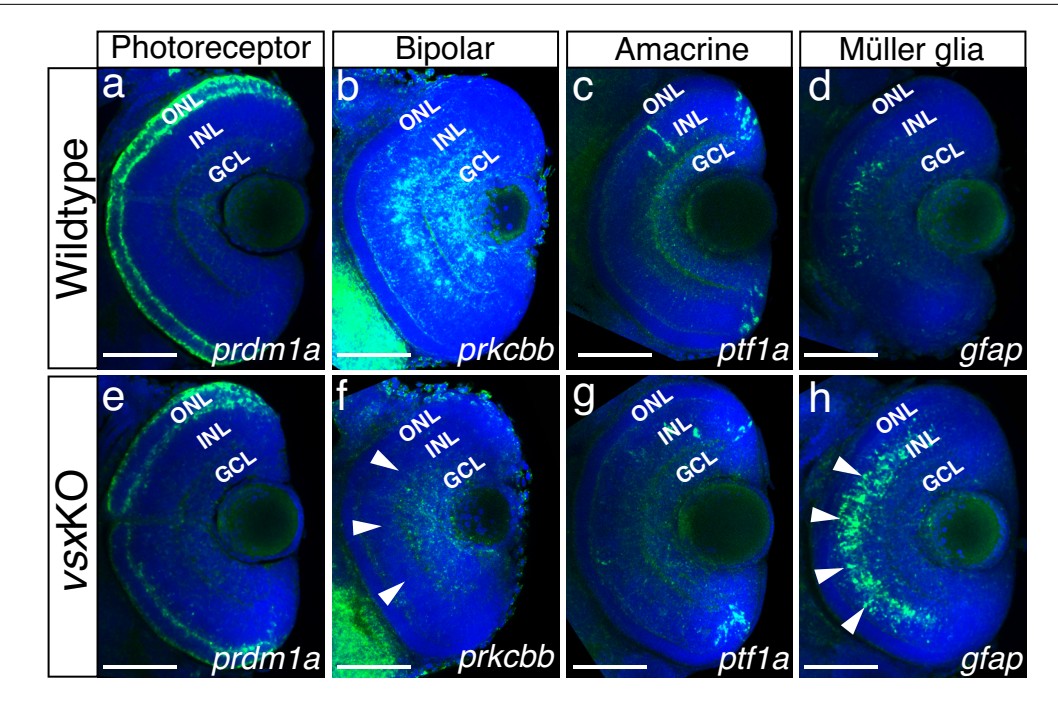

**Figure 4.** Altered expression of Bipolar and Müller glia cell markers in 3dpf *vsx* mutant fish. (**a-h**). Confocal sections from in toto *in situ* hybridization experiments using specific fluorescent probes to label different cell types in wildtype and *vsx*KO retinas at 72hpf. No clear differences in the expression of the photoreceptor marker *prdm1a* were observed in ONL of wildtype (**a**) and mutant samples (**e**). Bipolar cell marker *prkcbb* expression (**b, f**) is considerably reduced in the INL of *vsx*KO mutant retinas (**f**, white arrowheads) compared to wildtype (**b**). Similar expression of the amacrine cell marker *ptf1a* is observed in the INL from wildtype (**c**) and *vsx*KO (**g**) retinas. Increased expression of the Müller glia cell marker *gfap* (**d, h**) is observed in the INL of *vsx*KO samples (**h**, white arrowheads) compared to wildtype (**d**) retinas. ONL: outer nuclear layer, INL: inner nuclear layer, GCL: ganglion cell layer. Scale bar in (**a-h**): 50 µm.

The online version of this article includes the following figure supplement(s) for figure 4:

**Figure supplement 1.** Delayed photoreceptor differentiation is observed in *vsx*KO retinas.

**Figure supplement 2.** Analysis of INL markers *prox1, ptf1a,* and *pax6* in WT and *vsx*KO retinas.

**Figure supplement 3.** V2 spinal cord interneurons are not affected by the mutation of Vsx TFs.

program is delayed in the absence of *vsx* function and that cone cells are overrepresented in the ONL of the double mutants. A prolonged period of precursors' proliferation and/or competence could account for an increased number of PRs at larval stages, and thus for an expanded thickness of the ONL layer, as observed in double mutants at 6 dpf (*Figure 1*; *Figure 1—figure supplement 3*).

## INL/bipolar cells specification

In the zebrafish retina, *vsx1* and *vsx2* expression has been reported in complementary subsets of bipolar cells, with *vsx1* having a broader distribution and *vsx2* being restricted to a few bipolar subtypes (*Vitorino et al., 2009*). To analyse bipolar cell integrity in *vsx*KO embryos, we first performed immunohistochemistry for the general INL marker *prox1* (*Figure 4—figure supplement 2*; *Dyer, 2003*) and then fluorescent *in situ* hybridizations for the bipolar cell marker protein kinase Cb1 (*prkcbb*) (*Figure 4*). At 48hpf, no changes in the expression of *prox1* was detected between WT and *vsx*KO retinas (*Figure 4—figure supplement 2a, e*). However, at 72hpf the nuclear distribution of *prox1* in the INL is affected in *vsx*KO samples compared to WT retinas (*Figure 4—figure supplement 2b, b', f, f'*) suggesting a lack of bipolar cells in *vsx*KO retinas. This observation was further confirmed by the fact that at 72hpf *prkcbb* expression is very reduced, if not absent, in the INL of double mutant retinas compared to WT (n=5) (*Figure 4b and f*). These results are in agreement with our previous histological

(i.e., reduced INL thickness; *Figure 1—figure supplement 3*) and electrophysiological (i.e. reduced b-wave, *Figure 2*) observations in *vsx*KO larvae, and confirms that *vsx* genes are essential for bipolar cells specification in zebrafish. Although we can also detect a mild reduction of *prkcbb* at the GCL, and we cannot rule out transient defects in a particular RGCs subpopulation at 72 hpf (*Figure 4f*), the final RGC numbers seem normal in the *vsx*KO retinas as determined by DiI and DiO tracers (*Video 1*) as well as retinal histology at 6dpf (*Figure 1*).

## INL/amacrine cells (AC) specification

A detailed histological analysis of the INL architecture in wild type and *vsx*KO embryos suggested that ACs specification was not severely affected in the double mutant (*Figure 1d and h*). To confirm this point, we followed the expression of *ptf1a*, a transcription factor encoding gene that is expressed in horizontal and AC and has been shown to play an essential role in their specification in the mouse retina (*Fujitani et al., 2006*). Using a fluorescent probe against *ptf1a*, which is expressed transiently in all types of amacrine cells in the embryonic zebrafish retina (*Jusuf and Harris, 2009*), we could determine that the ACs differentiation wave progresses normally through the central retina in wild type and *vsx*KO embryos at 48 hpf (n=5) (*Figure 4—figure supplement 2c, g*). Later in development, at 72 hpf, *pft1a* expression was no longer detected in the central retina and appeared restricted to the most peripheral region, being expressed at similar levels in both wild type and *vsx*KO retinas (n=10) (*Figure 4c and g*). In addition, the expression of the differentiated amacrine cell marker *pax6* (*Hitchcock et al., 1996*) is not affected in *vsx*KO retinas compared to WT (*Figure 4—figure supplement 2d, h*). These observations suggest that *vsx* genes in zebrafish do not play a major role for amacrine cells specification.

## INL/Müller glia cell specification

The abnormal expression of *prox1* in the *vsx*KO fish INL (*Figure 4—figure supplement 2b', f'*) suggests an unbalance in the contribution of the different cell types present in that retinal layer. Müller glia cell bodies are located in the INL where they provide structural and functional support to the retinal neurons (*Goldman, 2014*). To investigate if their differentiation occurs normally in *vsx* double mutants, we used a *gfap* antisense probe as glial marker (*Bernardos and Raymond, 2006*). We found a clear increase in the expression of *gfap* in *vsx*KO retinas compared to WT (n=5) (*Figure 4d and h*), suggesting that this cell type is overrepresented in *vsx*KO retinas, which may compensate for the reduction in bipolar cells observed in these animals.

In addition to their expression in the retina, Vsx transcription factors are also expressed in spinal cord interneurons (V2a and V2b cells), which are important to coordinate motor neuron activity and locomotion (*Crone et al., 2008*; *Kimura et al., 2008*). As reported here, *vsx* double mutants die around 2 weeks post-fertilization. This lethality could be due by spinal cord interneuron specification defects that may restrict the movement of the animals. To examine the integrity of V2a and V2b interneurons, we label both cell types with *vsx1* and *tal1* fluorescent antisense probes, that are expressed in V2a and V2b neurons, respectively (*Figure 4—figure supplement 3*). No significant differences in V2a or V2b spinal cord interneurons density was observed between WT (n=6) and *vsx1Δ245-/-*, *vsx2Δ73-/-* mutant fish (n=8) at 24hpf (*Figure 4—figure supplement 3e*). These results suggest that V2 motoneurons are properly specified in *vsx*KO animals. In agreement with this observation, obvious swimming defects were not observed in *vsx*KO larvae.

## RNA-seq and ATAC-seq analyses of *vsx*KO reveal eye GRN robustness

The strong microphthalmia and abnormal specification of the neural retina reported in *vsx2* mutant mice (*Burmeister et al., 1996*; *Horsford et al., 2005*; *Rowan et al., 2004*) are in contrast to our observation that in *vsx*KO embryos/larvae optic cup identity is normally established and maintained. In *vsx*KO mutants, the morphology and size of the optic cup, the precursors' proliferation rate, as well as the distribution and expression of RPE specification markers (i.e. *tfec* and *bhlhe40*) appeared normal at 24hpf, as determined by PH3 staining and ISH, respectively (*Figure 3—figure supplement 1a-g*). The only parameter altered in *vsx*KO later in development is the onset of retinal differentiation, which appeared slightly delayed as determined by *atoh7* ISH at 26 hpf (*Figure 3—figure supplement 1h, i*). All these data point to a correct specification of the optic cup domains in the double mutants.

To gain insight into the molecular causes behind the discrepancy between mice and zebrafish mutants, we sought to investigate transcriptional and chromatin accessibility changes in mutant embryos during the specification of the neural retina (*Figure 5*). To this end, 18hpf embryo heads were collected from *vsx*KO and their wild type siblings and the rest of the tissue was used for PCR genotyping. We focused in this particular stage as it corresponds to the early bifurcation of the neural retina and RPE GRNs in zebrafish (*Buono et al., 2021*). To first identify changes in the *cis*-regulatory landscape associated to *vsx* loss of function, we examined wild type and mutant samples using ATAC-seq. This approach identified 1564 DNA regions with differential accessibility, most of them (1204) with a high fold change (i.e. log2 fold change > |1.5|). They include 633 regions more accessible in the mutant with a log2 fold change >1.5; and 571 less accessible with a log2 fold change <–1.5 (*Figure 5a and b*; *Figure 5—source data 1*). An analysis of enriched gene ontology terms for those genes (2219) neighboring the differentially opened regions revealed entries related to neuronal differentiation and eye development (*Figure 5—source data 2*). This observation suggests that *vsx* genes mutation results in the deregulation of hundreds of *cis*-regulatory elements mainly associated to retinal genes. In contrast, at a transcriptional level the comparative analysis of mutant and wild type samples by RNA-seq revealed expression changes only in a relatively small gene set (1018) (*Figure 5c*; *Figure 5—source data 3*). This collection comprised 41 up-regulated (log2 fold change >1.5) and 31 down-regulated (log2 Fold change <–1.5) genes, with only 3 up-regulated (*vsx1*, *znf1109,* and *znf1102*) and one down-regulated transcription factor (*znf1091*) above the threshold (log2 fold change > |1.5|) (*Figure 5c*). This observation indicated that the identified *cis*-regulatory changes are only translated in subtle changes at the transcriptional level. In fact, among the 2219 genes neighboring differentially open chromatin regions, only 5% (119) were associated to differentially expressed genes (*Figure 5d*). To further confirm the impact of *vsx* loss of function on the expression of core components of the neural retina GRN, we examined their levels by qPCR at 19 hpf (*Figure 5e*). Interestingly, in *vsx*KO embryos significant expression changes could be detected only for *rx2* and *lhx2b* (although below the threshold log2 fold change > |1.5|). In addition, the transcripts of the mutated genes *vsx1* and *vsx2* were significantly upregulated and downregulated respectively in mutant embryos at optic cup stages, as determined by qPCR at 19hpf (*Figure 5e*) and confirmed by ISH at 24hpf (*Figure 5—figure supplement 1*). Taken together, these analyses suggest that the general architecture of the retinal GRN was not significantly altered upon *vsx* genes mutation.

To further understand the phenotypic discrepancy observed between *vsx2* morphants and *vsx*KO mutants, we revisited previous splicing morpholino experiments (*Gago-Rodrigues et al., 2015*) confirming the reported microphthalmic phenotypes (*Barabino et al., 1997*; *Clark et al., 2008*; *Gago-Rodrigues et al., 2015*; *Vitorino et al., 2009*). Then we performed a full transcriptome analysis of *vsx2* morphants by RNA-seq at 18 hpf using embryo heads (*Figure 5—figure supplement 2a*). Principal components analysis (PCA) of wild type, morphant and double mutant datasets revealed a very different regulatory response between mutant and morphant samples (*Figure 5—figure supplement 2b*; *Figure 5—source data 4*). This differential behavior was evident when core components of the eye specification gene regulatory networks were examined: whereas mild transcriptional differences were detected for *rx2* and *lhx2b* in the mutants (*Figure 5*; *Figure 5—figure supplement 2c*), core components of the retinal network such as *rx3*, *rx1*, *six3b*, *vax2,* and *lhx2b* appeared upregulated in the morphants (*Figure 5—figure supplement 2c*). Particularly different was the expression of *vsx2* itself, which appeared strongly downregulated in *vsx*KO mutants, but strongly upregulated in *vsx2* morphants (*Figure 5—figure supplement 2d, f*). These results suggest that the dysregulation of the retinal network induced by the morpholinos (i.e. may be through compensatory mechanisms operating at the RNA level) is behind the early microphthalmia observed in the *vsx2* morphants.

Our results point at *vsx* genes having a different regulatory weight for neural retina specification/maintenance in different species. To gain insight into this hypothesis, we mutated two of the three paralogs (i.e. *vsx1* and *vsx2.2*) present in the genome of the far-related teleost medaka by CRISPR/Cas9 (*Figure 5—figure supplement 3*). We generated a 148 bp deletion in medaka *vsx1* deleting 29 bp of intron 2 and 119 bp of exon3 (*vsx1Δ148*). This frame shift mutation results in a deletion of 39 amino acids in the highly conserved DBD and the generation of a premature stop shortly after the deletion. In *vsx2.1*, a 319 bp deletion (*vsx2.1Δ319*) was generated encompassing intron2 (47 bp), exon3 (124 bp), and intron3 (147 bp). This frame shift mutation deletes 41 amino acids of the core DBD of the protein, mutate critical Arginines and generates a premature stop codon 66 amino acids

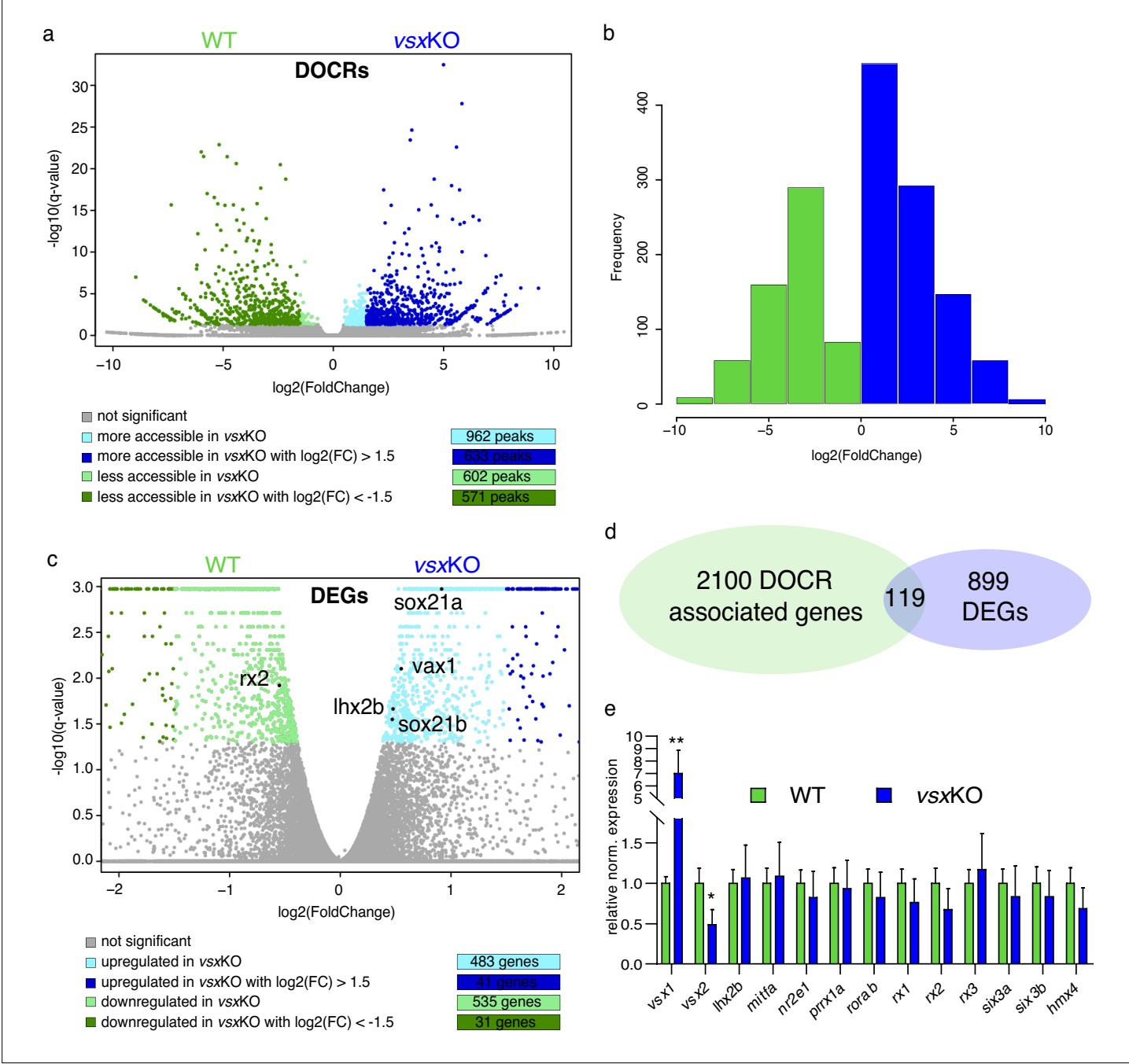

**Figure 5.** Lack of Vsx TFs in the forming retina is buffered by genetic redundancy. (**a**) Volcano plots illustrating chromatin accessibility variations upon *vsx1* and *vsx2* mutation in zebrafish retina at 18hpf. Each dot corresponds to an ATAC-seq peak, that is an open chromatin region. Grey dots indicate not significant variations, whereas colored dots point out significant differentially open chromatin regions. (**b**). Frequency of DOCRs' fold change values. (**c**). Transcriptome variations in *vsx*KO retina at 18hpf. The genes reported in the plot are the only known retinal regulators whose transcriptional levels are affected by the loss of Vsx factors, with a very modest fold change. Essentially, RNA-seq experiments did not highlight a remarkable change of the levels of the main TFs governing the retinal GRN. (**d**). Correspondence between genes associated with DOCRs from ATAC-seq and DEGs from RNA-seq. (**e**). qPCR of the main retinal TFs confirming the stability of the eye gene network expression after *vsx1* and *vsx2* loss (n=3). **p<0.001, *p<0.01 using one-way ANOVA. Data is shown as mean ± SD. DOCR: differentially open chromatin regions, DEG: differentially expressed genes.

The online version of this article includes the following source data and figure supplement(s) for figure 5:

**Source data 1.** List of all ATAC-seq peaks with differential accessibility in *vsx*KO vs WT.

**Source data 2.** Analysis of gene ontology terms for genes neighboring differentially opened regions in *vsx*KO.

*Figure 5 continued on next page*

Figure 5 continued

**Source data 3.** List of differentially expressed genes between WT and *vsx*KO embryos and cross-listing between DEGs and DOCRs from WT and *vsx*KO.

**Source data 4.** List of differentially expressed genes between WT vs *vsx2*MO and between *vsx*KO vs *vsx2*MO.

**Figure supplement 1.** The expression pattern of *vsx* mutant transcripts is misregulated in *vsx*KO animals during retina development.

**Figure supplement 2.** Transcriptomic divergence between *vsx*KO and *vsx2* morphant samples.

**Figure supplement 3.** Mutation of *vsx* genes in medaka impairs INL differentiation and eye growth.

**Figure supplement 4.** Normal eye size in *vsx*KO animals is observed at juvenile stages.

after the deletion. Interestingly, although the initial specification of the organ appeared normal also in medaka double mutant embryos at 4dpf, INL differentiation and eye growth was impaired at later stages in those animals (*Figure 5—figure supplement 3b–e*). This is in contrast to the normal eye size observed in *vsx*KO zebrafish larvae at 19dpf (*Figure 5—figure supplement 4*) and confirmed the assumption of differential regulatory weight among vertebrate species for *vsx* genes.

## Discussion

In this study, we explore the universality of Vsx functions in the development of the vertebrate eye, by generating CRISPR/Cas9 mutations of the 'visual system homeobox' genes *vsx1* and *vsx2* in the far related teleost models, zebrafish and medaka. Genetic analyses in the mouse, as well as the chick, had revealed two distinct functions for *Vsx* genes during eye development: an early requirement for proliferation and specification of the neural retina precursors, and a later role in the differentiation of bipolar neurons (*Burmeister et al., 1996*; *Horsford et al., 2005*; *Rowan et al., 2004*). These two developmental roles depend on consecutive waves of gene expression and thus can be uncoupled by genetic interference within specific developmental windows (*Goodson et al., 2020*; *Livne-Bar et al., 2006*). Moreover, in mice, Vsx biphasic activity follows a partially independent *cis*-regulatory control by enhancers active either in precursors, bipolar cells, or both (*Kim et al., 2008*; *Norrie et al., 2019*; *Rowan and Cepko, 2005*). Accordingly, CRISPR-mediated ablation of a distal bipolar enhancer results in the specific depletion of these cells, without leading to microphthalmia or compromising the early specification of the mouse retina (*Goodson et al., 2020*; *Norrie et al., 2019*).

Here, we show that *Vsx* activity is essential for bipolar cells differentiation in teleost fish, indicating a broadly conserved role for these genes across vertebrates. This observation suggests that the genetic program controlling bipolars specification was inherited from a common vertebrate ancestor. The fact that *Vsx* homologous genes are also expressed in the visual-system of the invertebrates *Drosophila* and Cuttlefish (*Sepia officinalis*) further suggests that the function for these homeobox genes in the specification of visual interneurons may be a common theme in all metazoans (*Erclik et al., 2008*; *Focareta et al., 2014*). The absence of an earlier eye phenotype in zebrafish *vsx*KO embryos allowed us examining in detail the consequences of *vsx* loss on cell fate determination and sight physiology. Both our histological and electrophysiological analyses confirmed bipolar cells depletion in *vsx*KO retinas. We show that, unable to acquire the bipolar fate, retinal precursors follow alternative differentiation trajectories such as undergoing apoptosis, extending their proliferative phase, or differentiating as photoreceptors or Müller glia cells. A detour toward photoreceptors fate in zebrafish is in agreement with previous studies in mice showing that the Blimp1/Vsx2 antagonism controls the balance between rods and bipolar cells (*Brzezinski et al., 2010*; *Goodson et al., 2020*; *Katoh et al., 2010*; *Wang et al., 2014*). Interestingly, in *vsx*KO retinas we observed a noticeable delay in the onset of cones and rods terminal differentiation markers, *zpr-1* and *zpr-3* respectively, indicating that Vsx activity is not only required for correct fate specification, but also to determine the timing of the differentiation sequence, in agreement with previous data in mice (*Rutherford et al., 2004*). Arguably, more intriguing was our observation of an increased number of Müller glia cells in *vsx*KO retinas. Both glial and bipolars cells are late-born retinal types deriving from a common pool of precursors with restricted developmental potential (*Bassett and Wallace, 2012*; *Hatakeyama et al., 2001*; *Rowan and Cepko, 2004*; *Satow et al., 2001*). In mice, however, a significant increase in Müller glia cells has not been reported in experiments genetically interfering with *vsx* either postnatally (*Goodson et al., 2020*; *Livne-Bar et al., 2006*) or specifically in bipolar cells (*Norrie et al., 2019*). This apparent

discrepancy might indicate some variations in the cell fate specification mechanisms among vertebrate species. Alternatively, the increase may have been overlooked in previous studies due to the small size of the Müller glia cell population. The fact that a trend toward an increase in Müller glia has been reported (*Livne-Bar et al., 2006*) may support this second possibility.

Despite the severe visual impairment and retinal lamination defects we observed in *vsx*KO larvae, their eyes appear normal in shape and size and no early morphological defects are observed in the optic cup. More importantly, neuro-retinal identity seemed perfectly maintained in double mutant animals, and we did not detect any trans-differentiation of the retina into pigmented cells. This finding, which is in contrast with the microphthalmia and the neural retina specification defects observed in mice (*Burmeister et al., 1996*; *Horsford et al., 2005*; *Rowan et al., 2004*), may indicate that *Vsx* genes do not play an early role in the establishment and maintenance of the neural retina identity in zebrafish. Although a potential rescue by maternally provided *vsx* genes could be hypothesized as an explanation for normal specification of the retina, this is an unlikely possibility as both transcripts are not detectable before zygotic genome activation (*White et al., 2017*). Our observations are also in contrast to previous reports in zebrafish using antisense oligonucleotides or morpholinos against *vsx2*, which show microphthalmia and optic cup malformations (*Barabino et al., 1997*; *Clark et al., 2008*; *Gago-Rodrigues et al., 2015*; *Vitorino et al., 2009*). A poor resemblance between morpholino-induced and mutant phenotypes has been previously described in zebrafish, with many mutations lacking observable phenotypes (*Kok et al., 2015*). Genetic compensation and, in particular, transcriptional adaptation (i.e. up-regulation of genes displaying sequence similarity) has been identified as the molecular mechanism accounting for genetic robustness in a number of these mutations (*El-Brolosy et al., 2019*; *Rossi et al., 2015*). However, our comparative transcriptomic analysis of *vsx*KO vs WT embryos does not support genetic compensation acting as a relevant mechanism at optic cup stages. We show that, despite that *vsx* loss of function results in the deregulation of hundreds of *cis*-regulatory regions associated to retinal genes, this has little impact on the expression of core components of the neural retina specification network.

Our previous analysis of transcriptome dynamics and chromatin accessibility in segregating NR/RPE populations indicated that the regulatory networks involved in the specification of the zebrafish eye are remarkably robust (*Buono et al., 2021*). In that study, we showed that the consensus motif 5'-TAATT-3', which is central to the neural retina *cis*-regulatory logic, is shared by many homeodomain TFs co-expressed during retinal specification; including not only *vsx1* and *vsx2*, but also *rx1*, *rx2*, *rx3*, *lhx2b*, *lhx9*, *hmx1*, and *hmx4*. Moreover, we show evidence that these TFs may co-regulate the same genes and cooperate within the same *cis*-regulatory modules (*Buono et al., 2021*). According to these observations, gene redundancy appears as a more parsimonious explanation for the absence of an early phenotype in *vsx*KO embryos. This would suggest that the regulatory weight of *vsx* genes within the retina network varies across vertebrate species. Several lines of evidence support this view. (i) Other mutations in genes encoding for TFs targeting the motif 5'-TAATT-3', such as *rx2* in medaka (*Reinhardt et al., 2015*) or *lhx2* in zebrafish (*Seth et al., 2006*) do not compromise the identity of the neural retina tissue either. (ii) Even in *vsx1/vsx2* double mutant mice, the central retina keeps the potential for differentiation into several neuronal types, indicating that other genes must cooperate in the specification of this tissue (*Clark et al., 2008*). In such scenario of complex epistatic interactions, it is not surprising the intrinsic variability in expressivity traditionally observed in ocular retardation mutants (*Osipov and Vakhrusheva, 1983*). In line with this, in *Vsx2* mutant mice has been shown that neural retinal identity defects and microphthalmia (but not bipolar cells differentiation) can be partially restored by simply deleting a cell cycle gene (*Green et al., 2003*). (iii) Finally, here we show that the mutation of the paralogs *vsx*1 and *vsx2.2* results in severe microphthalmia in medaka larvae. This finding confirms a variable role across species for *Vsx* genes in the specification and maintenance of the neural retina domain in vertebrates.

## Methods
### Animal experimentation and strains
All experiments performed in this work comply European Community standards for the use of animals in experimentation and were approved by ethical committees from Universidad Pablo de Olavide (#02/04/2018/041), Consejo Superior de Investigaciones Científicas (CSIC), the Andalusian

government and Universidad Mayor (#25/2018). Zebrafish AB/Tübingen (AB/TU) and medaka iCab wild-type strains were staged, maintained and bred under standard conditions (*Iwamatsu, 2004*; *Kimmel et al., 1995*). Zebrafish Vsx mutants were maintained harboring a single copy of *vsx1* (*vsx1Δ245+/-, vsx2Δ73-/-*) or *vsx2* (*vsx1Δ245-/-; vsx2Δ73+/-*), while medaka Vsx mutants were maintained in heterozygosis (*vsx1Δ148+/-, vsx2.1Δ319+/-*).

## Gene editing

Single guide RNAs (sgRNAs) targeting the DNA binding domains of *vsx1* and *vsx2* from zebrafish and *vsx1* and *vsx2.1* from medaka were designed using the CRISPRscan (*Moreno-Mateos et al., 2015*) and CCTop (*Stemmer et al., 2015*) design tools. Primers for sgRNA generation (see *Table 1*), were aligned by PCR to a universal CRISPR primer and the PCR product was further purified and used as template to sgRNA synthesis (*Vejnar et al., 2016*). To target individual *vsx* genes, a solution containing two sgRNAs (40 ng/μL each) and Cas9 protein (250 ng/μL) (Addgene; 47327) (*Gagnon et al., 2014*) were injected into one-cell-stage zebrafish and medaka embryos. Oligos used for screening of genomic DNA deletions flanking CRISPR target sites are detailed in *Table 1*. Wild-type and mutant PCR products from F1 embryos were further analysed by standard sequencing to determine germline mutations (Stab Vida).

## Histology

Zebrafish and medaka samples from different developmental stages harboring mutations in *vsx* genes were deeply anesthetized for 5–10 min with 160 mg/L of tricaine (ethyl 3-aminobenzoate methanesulfonate salt; MS-222; Merck) before dissecting their heads. Heads including both eyes were fixed in 4% w/v paraformaldehyde (PFA, Merck) in 0.1 M phosphate buffer overnight at 4 °C and the remaining tissue were kept for genotyping by conventional PCR. Wild-type and *vsx*KO sorted heads were then washed several times in 1 X PBS, incubated in 30% sucrose-PBS overnight at 4 °C, embedded in OCT (Tissue Tek) using cryomolds (Tissue Tek) and frozen in liquid nitrogen for short term storage at –80 °C. Cryosectioning of samples was performed using a Leica CM1850 cryostat and 20-μm-thick transverse sections were collected in glass slides (Super Frost Ultra Plus, #11976299, Thermo Fisher Scientific) for Phalloidin (#A12379, Alexa fluor 488, Thermo Fisher Scientific) and 4',6-Diamidine-2'-phenylindole dihydrochloride (DAPI, #10236276001, Merck) staining. Briefly, zebrafish and medaka eye transverse cryosections were dried at room temperature for ≥3 hr and washed with filtered PBST (0.1% Triton in 1 X PBS) five times for 5 min each wash. Then, slides were incubated with a solution containing 1/50 phalloidin Alexa fluor 488 in PBST supplemented with 5% DMSO (Merck) and covered with parafilm (Bemis) in a dark humid chamber overnight at 4 °C. After 30–60 min at room temperature, sections were incubated in a DAPI solution (1:1000 in PBST) and then washed five times for 5 min each wash with PBST. Slides were mounted with a drop of 15% glycerol in PBS and covered with 22x60 mm coverslips. Mounted slides were kept in the dark and confocal images were captured immediately (≤24 hr) using a Leica SPE microscope to detect Alexa 488 and DAPI signals from retina samples.

## Eye size and retina layer width measurements

Zebrafish embryos obtained from in-crosses of either *vsx1Δ245+/-, vsx2Δ73-/-* or *vsx1Δ245-/-; vsx2Δ73+/-*-fish, were raised for 2 weeks under standard conditions (*Kimmel et al., 1995*). At this stage larvae were anesthetized, the antero-posterior length was measured (in millimetres) and a lateral image of the head region was obtained (Olympus SZX16 binocular scope connected to an Olympus DP71 camera). In parallel, a tip of the tail was collected using a scalpel to extract genomic DNA using Chelex resin (C7901, Sigma) for PCR screening. Head images (all taken at the same magnification) sorted by their genotype (either wild-type or *vsx*KO) were analysed using Fiji software to measure eye surface. Total eye area was divided by fish antero-posterior length for each animal to normalize eye size. To measure retina INL and ONL layers width in zebrafish larvae, confocal images of eye cryostat sections from previously genotyped wild-type and *vsx*KO animals were taken using an immersion oil ×40 objective (SPE, Leica). These images were then analysed using Fiji software to measure INL and ONL width (μm).

**Table 1.** Nucleotide sequence of oligos used in this work.

Organism, gene of interest, application and nucleotide sequence is described in each column. Note that the target site is bolded in CRISPR/Cas9 primers used for vsx disruption.

| Organism | Gene | Application | Oligo sequence (5'–3') |
|---|---|---|---|
| Danio rerio | vsx1 | CRISPR/Cas9 | TAATACGACTCACTATA**GGGGTTCCTCAAGTTGATGGG**GTTTTAGAGCTAGAA |
| Danio rerio | vsx1 | CRISPR/Cas9 | TAATACGACTCACTATA**GGTTTACGCGAGAGAAATGC**GTTTTAGAGCTAGAA |
| Danio rerio | vsx2 | CRISPR/Cas9 | TAATACGACTCACTATA**GGTGCCGGAGGACAGAATAC**GTTTTAGAGCTAGAA |
| Danio rerio | vsx2 | CRISPR/Cas9 | TAATACGACTCACTATA**GGTGGAGAAAGCTTTTAACG**GTTTTAGAGCTAGAA |
| Danio rerio | vsx1 | Genotyping Fw | ATGACTGCCTTTCCGGTGAT |
| Danio rerio | vsx1 | Genotyping Rv | CTGCTGGCTCACCTAGAAGC |
| Danio rerio | vsx2 | Genotyping Fw | TCGTAATCTTTCCACTGATTCTGAT |
| Danio rerio | vsx2 | Genotyping Rv | TGTTCTAGAGCATATTGTCTGTTCC |
| Danio rerio | vsx1 | Cloning Fw | CGGGAAGAGAAGAAGCTACAGAT |
| Danio rerio | vsx1 | Cloning Rv | GCCTTCTCTTTTCCTCTTTTGA |
| Danio rerio | vsx2 | Cloning Fw | CTGTTTTGTCGGAAAGTTTGAA |
| Danio rerio | vsx2 | Cloning Rv | CCAGCTGGTAAGATGTAAATATTGTT |
| Danio rerio | ptf1a | Cloning Fw | GGCTTAGACTCTTTCTCCTCCTC |
| Danio rerio | ptf1a | Cloning Rv | CGTAGTCTGGGTCATTTGGAGAT |
| Danio rerio | gfap | Cloning Fw | GTTCCTTCTCATCCTACCGAAAG |
| Danio rerio | gfap | Cloning Rv | GATCAGCAAACTTTGAGCGATAC |
| Danio rerio | pkcb1 | Cloning Fw | GCGCAGTAAGCACAAGTTCAAGG |
| Danio rerio | pkcb1 | Cloning Rv | CCCAGCCAGCATCTCATATAGC |
| Danio rerio | prdm1a | Cloning Fw | TCAAAACGGCATGAACATCTATT |
| Danio rerio | prdm1a | Cloning Rv | AGGGGTTTGTCTTTCAGAGAAGT |

*Table 1 continued on next page*

*Table 1 continued*

| Organism | Gene | Application | Oligo sequence (5'–3') |
|---|---|---|---|
| *Danio rerio* | *tal1* | Cloning Fw | AGTATGATTTGCTCATCCTCCAA |
| *Danio rerio* | *tal1* | Cloning Rv | TTTGTTTGTTTGCGCATTTAATA |
| *Danio rerio* | *tfec* | Cloning Fw | TATAAAGACCGGACGGGGACAAC |
| *Danio rerio* | *tfec* | Cloning Rv | CAGCTCCTGGATTCGTAGCTGGA |
| *Danio rerio* | *bhlhe40* | Cloning Fw | TTGCAAATCGGCGAACAGGG |
| *Danio rerio* | *bhlhe40* | Cloning Rv | GGAAACGTGCACGCAGTCG |
| *Danio rerio* | *eef1a1l1* | qPCR Fw | TCCACCGGTCACCTGATCTAC |
| *Danio rerio* | *eef1a1l1* | qPCR Rv | CAACACCCAGGCGTACTTGA |
| *Danio rerio* | *vsx1* | qPCR Fw | TCTAGGTGAGCCAGCAGGAAT |
| *Danio rerio* | *vsx1* | qPCR Rv | CCATGTCGTGTCGCTGTCTT |
| *Danio rerio* | *vsx2* | qPCR Fw | GGGATTAATTGGGCCTGGAGG |
| *Danio rerio* | *vsx2* | qPCR Rv | GCTGGCAGACTGGTTATGTTCC |
| *Danio rerio* | *six3a* | qPCR Fw | AAAAACAGGCTCCAGCATCAA |
| *Danio rerio* | *six3a* | qPCR Rv | AAGAATTGACGTGCCCGTGT |
| *Danio rerio* | *six3b* | qPCR Fw | TCCCCGTCGTTTTGTCTCTG |
| *Danio rerio* | *six3b* | qPCR Rv | AGAAGTTTAGGGTGGGCAGC |
| *Danio rerio* | *lhx2b* | qPCR Fw | AGGCAAGATTTCGGATCGCT |
| *Danio rerio* | *lhx2b* | qPCR Rv | TCTCTGCACCGAAAACCTGTA |
| *Danio rerio* | *mitfa* | qPCR Fw | CTGATGGCTTTCCAGTAGCAGA |
| *Danio rerio* | *mitfa* | qPCR Rv | GCTTTCAGGATGGTGCCTTT |
| *Danio rerio* | *nr2e1* | qPCR Fw | CAAATCTGGCACACAGGGCG |
| *Danio rerio* | *nr2e1* | qPCR Rv | CGACGAACCGTTCACCTCTT |

*Table 1 continued on next page*

*Table 1 continued*

| Organism | Gene | Application | Oligo sequence (5'–3') |
|---|---|---|---|
| Danio rerio | prrx1a | qPCR Fw | CTCACCGTCATACAGTGCCA |
| Danio rerio | prrx1a | qPCR Rv | AGAGTCTTTGACAGCCCAGC |
| Danio rerio | rorab | qPCR Fw | ACAAACCAGCACCAGTTCGG |
| Danio rerio | rorab | qPCR Rv | CCTCCTGAAGAAACCCTTGCAT |
| Danio rerio | rx1 | qPCR Fw | AAGAACTTGCATCGGACGGT |
| Danio rerio | rx1 | qPCR Rv | TCGGAAGCTTGCATCCAGTT |
| Danio rerio | rx2 | qPCR Fw | TCGGGACGCATAAAGTGGAC |
| Danio rerio | rx2 | qPCR Rv | CGGGTCTCCCAAATCTGCAT |
| Danio rerio | rx3 | qPCR Fw | CCGAGTACAGGTGTGGTTCC |
| Danio rerio | rx3 | qPCR Rv | GTCAACCAGGGCTCTAACGG |
| Danio rerio | hmx4 | qPCR Fw | TGTCGACCCGCTTCTTTGAA |
| Danio rerio | hmx4 | qPCR Rv | TGATGAAGACAGCCATCCCG |
| Oryzias latipes | vsx1 | CRISPR/Cas9 | TAATACGACTCACTATA**GGCAGAGTGAGGTTCAGTGGG**GTTTTAGAGCTAGAA |
| Oryzias latipes | vsx1 | CRISPR/Cas9 | TAATACGACTCACTATA**GGTAGGGGCCTGACCTGGATT**GTTTTAGAGCTAGAA |
| Oryzias latipes | vsx2.1 | CRISPR/Cas9 | TAATACGACTCACTATA**GGGGATGATGAGAGTCAAGG**GTTTTAGAGCTAGAA |
| Oryzias latipes | vsx2.1 | CRISPR/Cas9 | TAATACGACTCACTATA**GGAAAAAATAACAGAATTGAG**TTTTTAGAGCTAGAA |
| Oryzias latipes | vsx1 | Genotyping Fw | AACAATAATTTAAAATGCGGAAAAA |
| Oryzias latipes | vsx1 | Genotyping Rv | GAAACTAAAATCCCATTCAGTGCT |
| Oryzias latipes | vsx2.1 | Genotyping Fw | ATATCACGGGAAATTAAAAATGCTC |
| Oryzias latipes | vsx2.1 | Genotyping Rv | AAGTCAAATGTGCCATTGTTAGTC |

## Electroretinography (ERG)

ERG was recorded on 5 dpf larvae as previously described (*Zang et al., 2015*). 100ms light stimuli delivered by HPX-2000 (Ocean Optics) were attenuated (log-4 to log0) by neutral density filters and given with an interval of 15 s. Full light intensity was measured by spectrometer (Ocean Optics, USB2000+) with spectrum shown in S1 (SpectraSuite, Ocean Optics). Electronic signals were amplified 1000 times by a pre-amplifier (P55 A.C. Preamplifier, Astro-Med. Inc, Grass Technology), digitized by DAQ Board (SCC-68, National Instruments) and recorded by self-written Labview program (National Instruments). Figures were prepared using Microsoft Excel 2016.

## Optokinetic response (OKR)

The OKR was recorded by the experiment setup as previously described (*Mueller and Neuhauss, 2010*). Briefly, 5dpf larvae were stimulated binocularly with sinusoidal gratings. To determine the contrast sensitivity, a spatial frequency of 20 cycles/360° and an angular velocity of 7.5 deg/s were used with varying contrast (5, 10, 20, 40, 70, and 100%). To study the spatial sensitivity, an angular velocity of 7.5 /s and 70% of the maximum contrast was used with different spatial frequency (7, 14, 21, 28, 42, 56 cycles/360°). To analyse the temporal sensitivity, maximum contrast and a spatial frequency of 20 cycles/360° were applied with increasing temporal frequency (5, 10, 15, 20, 25, 30 deg/s). Figures were presented by SPSS (Version 23.0. Armonk, NY: IBM Corp).

## Immunohistochemistry in sections

Zebrafish wild-type and *vsx*KO retina sections from different developmental stages were analysed for the detection of apoptotic and mitotic cells using rabbit anti-active caspase-3 antibodies (BD Biosciences, 559565) and rabbit anti-phospho-histone H3 antibodies (Merck Millipore, 06–570), respectively. For the detection of cone and rod photoreceptors, zpr1 (ZIRC) and zpr3 (ZIRC) antibodies were used, respectively. Briefly, eye transverse cryosections were dried at room temperature for ≥3 hr, washed five times for 5 min each with PBST, blocked for ≥1 hr with 10% fetal bovine serum in PBST and incubated overnight in a humid chamber at 4 °C with the corresponding primary antibody. All primary antibodies were diluted 1:500 in blocking solution. After several washes with PBST, a 1:500 dilution of the secondary antibody (Alexa Fluor 555 goat anti-rabbit or goat anti-mouse antibodies, Thermo Fisher, #A-21429 and #A-21422, respectively) was added for 2 hr at room temperature. Following extensive washes with PBST, slides were mounted in 15% glycerol/PBS solution and sealed with 22x60 mm coverslips. Immunofluorescence confocal images were taken using a Leica SPE confocal microscope.

## Whole-mount embryo immunofluorescence

Embryos collected from in-crossed *vsx1*+/-; *vsx2*-/- adult fish were dechorionated and fixed at 72 hpf with 4% Formaldehyde in PBS (FA). Fixed embryos were washed with PBS-Tween 0.5%-Triton 0.5% (PBST), treated with Proteinase K (10 µg/mL in PBST) for 30 min at 37 °C followed by PBST washes and a post-fixation step in FA for 30 min at room temperature (RT). After PBST washes, embryos were treated with cold acetone at –20 °C for 20 min, then washed again with PBST and incubated with freshly prepared blocking solution (5% normal goat serum, 1% BSA, 1% DMSO in PBST) at RT for 2 hr. Primary antibody specific for zebrafish Prox1 (GeneTex, GTX128354) and Pax6 (GeneTex, GTX128843) were diluted 1:100 in blocking solution and embryos were incubated overnight (ON) at 4 °C. Embryos were subsequently washed with PBST and incubated ON at 4 °C in the dark with the Alexa FluorTM 488 Goat anti-rabbit antibody (Invitrogen), diluted 1:500. Finally, embryos were washed with PBST and incubated ON at 4 °C with DAPI (Sigma) diluted 1:5000 in PBST. For imaging, embryos were embedded in 1% low-melting point agarose, transferred to glass-bottom culture dishes (MatTek corporation) and manually oriented. Confocal laser scanning microscopy was performed using an LSM 880 microscope (Zeiss). Images were processed using Fiji. After imaging, embryos were genotyped by PCR to identify *vsx1*-/-; *vsx2*-/- double mutant embryos.

## RNA *in situ* hybridization

Fluorescence *in situ* hybridization experiments were performed as previously described (*Bogdanović et al., 2012*). Fragments of the *vsx1*, *vsx2*, *ptf1a*, *prdm1a*, *gfap*, *prkcbb*, *tfec*, *bhlhe40* and *tal1* genes were PCR amplified from zebrafish cDNA (SuperScript IV VILO Master Mix ThermoFisher Scientific, #11756050) using specific primers (*Table 1*). For *vsx1* and *vsx2* genes, the deleted region of

the coding sequence in *vsx*KO mutants was excluded from the amplified fragment. PCR products were cloned into StrataClone PCR Cloning vector (Agilent, #240205), linearized with XbaI restriction enzyme (Takara, #1093B) and transcribed with a DIG-labeling Kit (Roche, #11277073910) using T3 polymerase (Roche, #11031163001) to obtain digoxigenin-labeled antisense probes. Probes were used at a final concentration of 2 ng/µl diluted in hybridization buffer (*Thisse and Thisse, 2008*). For *atoh7*, a colorimetric antisense digoxigenin-labeled RNA probe was prepared as reported elsewhere (*Masai et al., 2000*).

## Morpholino injections

The *vsx2*E2I2 splicing morpholino was obtained from Gene Tools and injected as reported before (*Gago-Rodrigues et al., 2015*). For RNA-seq experiments, *vsx2* morphants where co-injected with lyn-Td-tomato mRNA at a concentration of 50 ng/µL. At 16 hpf, red fluorescent embryos were pooled under the stereoscope and heads were dissected at 18 hpf for total RNA extraction.

## RNA-seq

Total RNA was extracted from 18 hpf zebrafish embryos' heads using 1 ml TRIzol (Invitrogen, #15596026) following the manufacturer's protocol. The trunk and tail of the embryos was used to extract genomic DNA using Chelex resin (C7901, Sigma) for PCR screening. Potential DNA contamination was eliminated by treating RNA samples with TURBO DNAse-free kit (Ambion, #AM1907). The concentration of the RNA samples was evaluated by Qubit spectrophotometer (Thermo Fisher). Libraries were prepared with TruSeq stranded mRNA kit (Illumina) and sequenced 2x125 bp on an Illumina Nextseq platform. We obtain at least 33 million reads per sample. Three biological replicates were used for each analysed condition. Reads were aligned to the *danRer10* zebrafish genome assembly using hisat2 (*Kim et al., 2015*). Transcript abundance was estimated with Cufflinks v2.2.1. Differential gene expression analysis was performed using Cuffdiff v2.2.1, setting an adjusted P-value <0.05. PCA analysis were done using CummeRbund, R package version 2.40.0 (*Goff and Trapnell, 2022*). Heatmap visualization was obtained with Clustvis (*Metsalu and Vilo, 2015*) using the FPKMs normalized by row as input.

## qPCR

cDNA retrotranscription and qPCR were performed as previously described (*Vázquez-Marín et al., 2019*). *eef1a1l1* gene was used as housekeeping normalization control. Primer sequences for amplified genes are detailed in *Table 1*.

## ATAC-seq

Each ATAC-seq sample was obtained starting from a single 18 hpf zebrafish embryo's head manually dissected, while the trunk and tail was kept for conventional PCR genotyping. Tagmentation and library amplification were performed using the FAST-ATAC protocol previously described (*Corces et al., 2016*). We obtained at least 70 M reads from the sequencing of each library. For data comparison, we used two biological replicates for each condition. Reads were aligned to the danRer10 zebrafish genome using Bowtie2 (*Langmead and Salzberg, 2012*) with -X 2000—no-mixed—no-unal parameters. PCR artifacts and duplicates were removed with the tool rmdup, available in the Samtools toolkit (*Li et al., 2009*). In order to detect the exact position where the transposase binds to the DNA, read start sites were offset by +4/–5 bp in the plus and minus strands. Read pairs that have an insert <130 bp were selected as nucleosome-free reads. Differential chromatin accessibility was calculated as reported (*Magri et al., 2019*). All chromatin regions reporting differential accessibility with an adjusted p-value <0.05 were considered as DOCRs. All the DOCRS have been associated with genes using the online tool GREAT (*McLean et al., 2010*) with the option 'basal plus extension'. Gene ontology enrichment analysis of the genes associated with DOCRs was performed with PANTHER (*Mi et al., 2021*).

## Labeling of retinotectal projections (DiI/DiO injections)

Following PCR genotyping, 6 dpf wild-type and *vsx* mutant larvae were fixed in 4% PFA overnight, washed in PBS and embedded in 1% low melting agarose (Sigma, #A9414) in PBS on an agarose filled Petri dish for injection. Each eye (between the lens and the retina) of the fish was injected either with

1% DiI (Invitrogen, #D275) or 1% DiO (Invitrogen, #D282) solutions in Chloroform (or dimethylformamide) with a pulled capillary glass mounted on a micromanipulator and under a stereomicroscope. A PV820 Pneumatic PicoPump (WPI) with the appropriate setting to deliver pressure to label the whole retina was used. Injected simples were washed in PBS, maintained overnight at 4 °C and mounted on low melting agarose to image on a Zeiss LSM 710 confocal microscope. Z-stacks (0.5 µm x 0.5 µm x 1 µm) were collected to visualize the optic nerve reaching the tectum and 3D reconstructions were generated using Zen blue edition software (Zeiss).

## Total protein extraction and western blotting analysis

Vsx1 and Vsx2 protein presence was analysed by Western blotting. To accomplish this, three different samples were prepared: the first two contained 20 heads of wildtype or *vsx*KO embryos at 24 hpf stage and the third sample comprised 20 wildtype embryos at 1.5 hpf. Each set of embryos were shaken for 5 min at 1100 rpm in deyolking buffer (55 mM NaCl, 1.8 mM KCl and 1.25 mM NaHCO$_3$). Tubes were then centrifuged at 300 g for 30 s, and subsequently the pellets were rinsed with wash buffer (10 mM Tris-HCl pH8.0, 110 mM NaCl, 3.5 mM KCl and 2.7 mM CaCl$_2$). Then, each pellet was resuspended in 25 µL SDS buffer (100 mM TrisHCl pH 6.8, 4% SDS, 20% glycerol and 200 mM DTT) and heated at 95 °C for 5 min. After that, samples were centrifuged at 16,000 *g* for 20 min at 4 °C and supernatants were collected. Protein extracts were loaded in 10% TGX Stain-FreeTM FastCastTM Acrylamide (BioRad) and blotted onto nitrocellulose membranes. Western blot normalization was performed using total protein load following manufacturer protocol for Stain Free gels (Bio Rad). Vsx1 (A10801, https://www.antibodies.com/) and Vsx2 (X1180P, Abintek) antibodies were used at a 1:500 dilution, followed by incubation with anti-Rabbit IgG-HRP secondary antibody (AP160P, Chemicon), diluted to 1:10000 for Vsx1 detection and Rabbit anti-Sheep IgG-HRP secondary antibody (402100, Calbiochem) diluted to 1:2000 for Vsx2. Chemiluminescent signals were detected with SuperSignal West Femto Substrate (Thermo Scientific) in a ChemiDoc MP Imaging System (BioRad).

## Statistical analysis

Quantitative data were evaluated using Prism 9.0 GraphPad software. Two-way ANOVA and a Tukey post hoc test was used to analyse ERG data, one-way ANOVA for OKR recordings and qPCR. Unpaired *t* test were used for PH3+ cell counts, C3+ cell counts, total eye area, retina layers' width, trunk V2 neuron comparisons and Zpr1/3 fluorescent intensity labeling. n values and significance levels are indicated in figure legends.

## Acknowledgements

We thank Marta Magri for their scientific advice and the CABD Proteomics, Aquatic Vertebrates and Functional Genomics facilities for their excellent technical assistance. This work was supported by grants awarded to JL from ANID (FONDECYT Iniciación #11180727, FONDECYT Regular #1230903) and JRM-M from Junta de Andalucía (Reference PY20_00006), CSIC (Reference 2020AEP014), and Spanish Ministry of Science, Innovation and Universities: (References BFU2017-86339P, RED2018-102553-T, PID2020-112566GB-I00, and CEX2020-001088-M).

## Additional information

### Funding

| Funder | Grant reference number | Author |
| --- | --- | --- |
| Fondo Nacional de Desarrollo Científico y Tecnológico | 11180727 | Joaquín Letelier |
| Fondo Nacional de Desarrollo Científico y Tecnológico | 1230903 | Joaquín Letelier |
| JUNTA DE ANDALUCIA | PY20_00006 | Juan R Martínez-Morales |

| Funder | Grant reference number | Author |
| --- | --- | --- |
| Consejo Superior de Investigaciones Científicas | 2020AEP014 | Juan R Martínez-Morales |
| Spanish Ministry of Science, Innovation and Universities | BFU2017-86339P | Juan R Martínez-Morales |
| Spanish Ministry of Science, Innovation and Universities | CEX2020-001088-M | Juan R Martínez-Morales |
| Spanish Ministry of Science, Innovation and Universities | PID2020-112566GB-I00 | Juan R Martínez-Morales |
| Spanish Ministry of Science, Innovation and Universities | RED2018-102553-T | Juan R Martínez-Morales |

The funders had no role in study design, data collection and interpretation, or the decision to submit the work for publication.

## Author contributions

Joaquín Letelier, Conceptualization, Data curation, Formal analysis, Supervision, Funding acquisition, Investigation, Methodology, Writing – original draft, Project administration, Writing – review and editing; Lorena Buono, Jorge Corbacho, Data curation, Formal analysis, Investigation, Methodology; María Almuedo-Castillo, Data curation, Formal analysis, Investigation; Jingjing Zang, Data curation, Investigation; Constanza Mounieres, Sergio González-Díaz, Ana Sousa-Ortega, Ruth Diez del Corral, Investigation; Rocío Polvillo, Stephan CF Neuhauss, Funding acquisition, Investigation; Estefanía Sanabria-Reinoso, Juan R Martínez-Morales, Conceptualization, Resources, Formal analysis, Supervision, Funding acquisition, Investigation, Writing – original draft, Project administration, Writing – review and editing

## Author ORCIDs

Joaquín Letelier http://orcid.org/0000-0002-2406-0337
Lorena Buono http://orcid.org/0000-0002-5457-4515
Ruth Diez del Corral http://orcid.org/0000-0003-2649-7214
Stephan CF Neuhauss http://orcid.org/0000-0002-9615-480X
Juan R Martínez-Morales http://orcid.org/0000-0002-4650-4293

## Ethics

All experiments performed in this work comply European Community standards for the use of animals in experimentation and were approved by ethical committees from Universidad Pablo de Olavide (#02/04/2018/041), Consejo Superior de Investigaciones Científicas (CSIC), the Andalusian government and Universidad Mayor (#25/2018). Zebrafish AB/Tübingen (AB/TU) and medaka iCab wild-type strains were staged, maintained and bred under standard conditions.

## Decision letter and Author response

Decision letter https://doi.org/10.7554/eLife.85594.sa1
Author response https://doi.org/10.7554/eLife.85594.sa2

# Additional files

## Supplementary files
• MDAR checklist

## Data availability
Sequencing data have been deposited in GEO under accession code GSE189739.

The following dataset was generated:

| Author(s) | Year | Dataset title | Dataset URL | Database and Identifier |
|---|---|---|---|---|
| Letelier J, Buono L | 2022 | Mutation of Vsx genes in zebrafish highlights the robustness of the retinal specification network | https://www.ncbi.nlm.nih.gov/geo/query/acc.cgi?acc=GSE189739 | NCBI Gene Expression Omnibus, GSE189739 |

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
