## [Editor Report]

This study provides important insights into how tissue specification networks, while often employing conserved genes across species, can differ in their network architecture, resulting in differences in how they buffer perturbations. This is shown for the Visual System Homeobox genes (VSX) in the zebrafish retinal specification pathway, where yet-to-be-defined compensatory mechanisms prevent microphthalmia in the absence of VSX function, something not observed in humans or mice. The evidence supporting the conclusions of the study is solid and provides a foundation for further molecular and genetic analysis of retinal specification. This work is relevant to developmental biologists with interests in tissue specification and gene regulatory networks.

---

## [Decision Letter]

**Decision letter after peer review:**

[Editors’ note: the authors submitted for reconsideration following the decision after peer review. What follows is the decision letter after the first round of review.]

Thank you for submitting the paper "Mutation of Vsx genes in zebrafish highlights the robustness of the retinal specification network." for consideration by *eLife*. Your article has been reviewed by 3 peer reviewers, and the evaluation has been overseen by a Reviewing Editor and a Senior Editor. The following individual involved in review of your submission has agreed to reveal their identity: William A Harris (Reviewer #1).

Comments to the Authors:

While you can see that there was some enthusiasm for your work and all three reviewers agreed that this is an interesting study, too many questions remained open at this point to move forward with the current submission. I and all reviewers feel that these revisions would be a substantial amount of work and you might want to consider another journal. If however, all experimental needs could be met, a submission of a reworked submission might be considered.

*Reviewer #1 (Recommendations for the authors):*

This is a solid paper that makes a contribution to understanding the roles of Vsx genes in retinal development in vertebrates. There are often 2 or more Vsx genes in different vertebrates so the effects of knocking them out singly can only partially reveal the function of a full Vsx knockout. This manuscript shows that a CRISPR KO of the both Vsx genes in zebrafish leads to retinal phenotypes that are similar in several ways to that of Vsx2 mutant (ocular retardation or) mice, i.e. lack of bipolar cells accompanied fate switching, altered proliferation, and increased cell death.

There is also a significant difference between VsxKO zebrafish and or mutant mice. The KO fish do not appear to show microphthalmia while or mutant mice do. To begin to explore this, the authors looked at how Vsx2KO affects other genes in involved in retinal development in zebrafish using sequence analyses and gene ontologies, and they found that although the VsxKO impinges on the regulation of thousands of genes, very few actually showed significant changes at the transcriptome level. Most dysregulated, as would have been expected, were the mutant transcripts coming off the CRISPRed Vsx genes themselves. Although there were also changes in the level of Rx (as has previously reported in Vsx2 MO fish). Interestingly Rx mutants and Vsx2MO fish show microphthalmia.

It would have been interesting to compare how the GRN in zebrafish in affected by Vsx2KO to that of the or mutant mice, or even Vsx2 morphants. A difference might reveal why the one shows microphthalmia and the other not.

Is it possible to get transcriptomic data from or mutant mice or Vsx2 morphant fish for comparison of the genetic robustness and clues to the or phenotype and why it isn't seen in Vsx2KO?

I would have liked to have seen the INL in better detail, for example using Vsx reporter lines. So one could see what happens to the fate switched bipolar cells.

*Reviewer #2 (Recommendations for the authors):*

In Letelier et al. the authors investigate the functional conservation of the visual system homeobox transcription factors vsx1 and vsx2 using zebrafish as a model for the teleost species. Genetic mutations were generated for each gene through CRISPR-Cas9 targeting of the conserved DNA binding domain, which resulted in a vsx1 frameshift mutation and vsx2 early stop codon. Individual vsx1 or vsx2 homozygous mutants are viable and only vsx1 -/- animals present ocular phenotypes, specifically a decrease in visual function based on electroretinograms, the visual background adaptation reflex, and optokinetic repsonse. The lack of phenotypes, such as small eyes, in vsx2-/- animals is a departure from mammalian models of vsx2 loss of function. The vsx1-/-;vsx2-/- animals also do not feature small eyes but do present other retinal developmental phenotypes. These include increased outer nuclear layer thickness, decreased inner nuclear layer thickness, visual defects, increased cell death, prolonged proliferation, delayed photoreceptor differentiation, and changes to various cell fates. Finally, ATACseq analysis of open chromatin in vsx1-/-;vsx2-/- animals revealed significant changes to open chromatin while RNAseq only resulted in minor changes to RNA transcripts. Based on the collection of results, the authors conclude early zebrafish visual system development differs from mammals due to compensatory mechanisms built into the eye field gene regulatory network.

The strengths of this manuscript include the high quality data presented and the variety of techniques used to investigate the changes to vision and retinal histogenesis in vsx1;vsx2 double homozygous mutants. Despite these strengths there are other mechanistic explanations for the differences between zebrafish and mammalian vsx1;vsx2 function, which can be addressed with additional analysis.

1. Due to the lethality of vsx1-/-;vsx2-/- animals, the only way to produce double homozygous offspring is to use parents with a wild type allele of either gene. This leads to the potential of maternal contributions of the wild type allele during early development, including the time window of optic cup formation. A maternal contribution of either vsx1 or vsx2 could contribute to the lack of early phenotypes, such as small eyes. The lack of microphthalmia in the mutants studied may be due to the maternal contribution of functional vsx1 or vsx2 genes depending on the breeding scheme.

2. The vsx1 mutation is an in frame deletion and despite the loss of the DNA binding domain, several other regions of conserved sequence are retained. In addition, vsx1 is one of three genes significantly upregulated in the RNAseq analysis with a log2 fold change > |1.5|in vsx1-/-;vsx2-/- animals. The qPCR analysis of eye related genes resulted in a 7 fold increase in vsx1. This large increase in the mutant form of vsx1 may be masking phenotypes or inducing different phenotypes than true loss of function mutations.

3. The authors show the ONL of vsx1-/-;vsx2-/- retinas are larger compared to controls, yet there are no clear changes to the photoreceptor precursor marker prdm1a (Figure 4). Further, even though there is a delay in photoreceptor differentiation there are no visible differences in expression of Zpr1+ cones and Zpr3+ rods (Figure S4). It is unclear based on these set of results to what is causing the increased size of the ONL in vsx1-/-;vsx2-/- retinas.

4. The authors should address the issue of maternal contribution of wild type vsx1 and vsx2. This can be difficult to do experimentally, especially due to the mutations not completely eliminating vsx1 and vsx2 through nonsense mediated decay. If it is not possible to assess maternal wild type mRNA or protein the authors should comment on this issue within the manuscript text.

5. In relation to point 1, while there are minimal changes to the transcriptome at 18/19hpf there are changes to bipolar and MG cells, suggesting late transcriptional changes in the mutants. Does this provide further evidence for maternal contribution? Why does genetic redundancy buffer the early but not late phenotypes? Is otx2a or otx2b expression changed in the mutants?

5. Overexpression experiments with the mutant form of vsx1 should be performed to address whether or not it produces a retinal phenotype similar to what is observed in the mutants. A large presence of the Vsx1 mutant protein may produce ectopic interactions otherwise not observed.

6. Another interesting observation from the study is the increase in MG cells based on GFAP staining. A previous study from Hatakeyama et al. (2001 Development 128 (8): 1313-1322.) found misexpression of Vsx2 (Chx10) in explant cultures increased the numbers of MG cells. Livine-Bar et al. (2006. PNAS 103 (13) 4988-4993) performed a similar experiment and although they didn't report a significant increase in MG cells, the bar graph containing cell counts does show a trend towards increased MG (Figure 1D). Is there a role for the truncated version of the mutant Vsx2 generated in this study? While vsx2 levels are decreased via qPCR in the double mutants, the in situ in Figure S8 shows a significant increase in vsx2 expression at 72 hpf.

7. Cell counts of rods and cones will help clarify the disparity in staining abundance and ONL thickness changes. There is precedent for an increase in rods with late knockdown of Vsx2 expression (Livine-Bar et al. 2006).

*Reviewer #3 (Recommendations for the authors):*

Letelier and colleagues examined the genetic requirements of the two paralogous VSX genes, vsx1 and 2, in zebrafish retinal development by generating Crispr deletions that target each gene. A single copy of either gene is sufficient to prevent macroscopic changes or lethality, but double mutants (vsxKO) die at approximately two weeks. Most of the study then centers on the retinal phenotypes of the single and vsxKO phenotypes, largely focusing on the known phenotypic characteristics of vsx1 and vsx2 mutants or knockdowns in other species such as mouse, medaka, and chick. Notable differences in zebrafish versus the other species were the lack of microphthalmia, retinal hypocellularity in the early retina, and ectopic pigmentation. These phenotypes are typically due to vsx2 loss of function, and the authors show that genetic compensation or redundancy between vsx1 and vsx2 is not the reason. A similar lack of compensation/redundancy between vsx1 and vsx2 for these early retinal phenotypes was previously shown in mouse and was suggested in zebrafish. Notable similarities were the reduction in bipolar cell function and the resulting defects in visual signaling as noted by ERG and the visual background adaptation (VBA) reflex. Other cross-species similarities were delayed differentiation of photoreceptors, and a skew in the proportions of late cell types, notably rods and Muller glia. Bulk RNA sequencing and ATAC seq in 18 hpf wild type and mutant retina demonstrated a limited degree of overlap with respect to differentially expressed genes and nearby differentially accessible chromatin regions. These last findings combined with the lack of microphthalmia and apparent lack of changes in early retinal development led the authors to suggest that these differences highlight the robustness of the retinal specification network.

With a couple of exceptions, the phenotypic characterizations and interpretations are supported with data, but the depth of analysis is generally not sufficient to reveal mechanistic insights or push the field beyond what has already been characterized. The main conclusion that the work provides insight into the robustness of the retinal specification network is not supported by data. The lack of an early retinal phenotype reported here is not consistent with other reports of vsx knockdown in zebrafish. The speculation that the differences are due to morpholinos versus genetic manipulation is concerning because no proof of this is provided. In that this outcome calls into question prior research by others and one of the senior authors of this study, it is imperative to provide an explanation supported by data.

1. The generation of the Crispr deletions is an important step for better understanding the genetic requirements of vsx1 and vsx2 in zebrafish. More information is needed, however, to understand the true nature of the mutations, This should include a graphic showing how the mutant alleles are altered with respect to open reading frames, domains, and structural motifs. Data should also be provided demonstrating the degree of mutant protein expression, preferably by western blot. This could then allow the authors to determine whether these alleles are nulls, hypomorphs, neomorphs, etc. This issue could also be relevant to understanding why the vsx2 mutant differs from morpholino knockdown

2. The lack of microphthalmia is an unexpected outcome. The apparent lack of a reduced proliferation phenotype at 48 hpf and a lack of overt pigmentation in the retina suggest that the early retinal phenotypes observed in vsx2 mutant mice, medaka, and vsx2(R200Q) mutant human organoids are not occurring in zebrafish. But the earliest stages of retinal development were not presented. It is well established that the proliferation and identity defects in vsx2 mutant mice are revealed very soon after vsx2 onset. In addition, the initiation of neurogenesis is delayed in mouse vsx2 mutants. If the authors performed an earlier phenotypic analysis and their interpretation holds, this would be one of the more novel aspects of the study. And the study would be greatly strengthened by data that provide new mechanistic insights and/or future directions for the field.

3. A lack of pigmentation does not rule out changes in gene expression that would indicate problems with retinal identity. Pigmentation is the outcome of a differentiation pathway, whereas identity issues could be indicated by the ectopic expression of genes related to other tissues such as RPE, ciliary epithelium, etc.

4. Issues 2 and 3 pertain to the early retinal phenotypes observed with multiple alleles in mice, medaka and to a certain extent with results in zebrafish vsx2/chx10 morphants. The apparent differences across species could very well be interesting, but the differences between prior morpholino data versus the mutant data here is troubling. First, Clark et al. (PMCID: PMC3315787) showed obvious changes in cyclind1 and vsx1 in chx10 morphants at 24 hpf, a finding very consistent across species. Second, this is concerning in its implication for one of the senior author's prior work. Gago-Rodrigues et al. (DOI: 10.1038/ncomms8054) performed well controlled experiments showing that their vsx2 knockdown was specific and used this approach to demonstrate microphthalmia and defective optic cup morphogenesis through a mechanism positing that vsx2 promotes the expression of ojoplano (opo). It was a rigorous study and very consistent with what the corresponding author's group reported in medaka. The lack of data to support a suitable explanation diminishes the findings presented here. If the authors believe it is due to redundancy by a non-vsx gene, then that should be shown.

5. The data for visual impairment and reduced bipolar function is strong. However, evidence of the fate of the bipolar cells is lacking. In germline vsx2 mouse mutants and early postnatal Crispr inactivations, the bipolar cells fail to form or rapidly die. In late postnatal Crispr inactivation and in vsx1 mutants, bipolar cells form but don't differentiate properly and their function is impaired. More direct evidence of how bipolar cell formation or function is affected should be provided.

6. Related to Point 2 above:

a. At the minimum, a 24 hpf timepoint should be analyzed with proliferation and cell cycle markers (e.g. pHH3, cyclind1 BrdU, pcna) and expression for RPE/pigmentation genes (e.g. mitf, otx1, dct, to name a few) should be done.

b. The delay of neurogenesis in mouse vsx2 mutants is clearly indicated by markers of retinal ganglion cells, Tuj1 staining, and markers neurogenic progenitors (e.g. atoh7, neurog2, ascl1, otx2 (also marks photoreceptor precursors)). These or equivalent zebrafish markers should be assessed at the appropriate timepoint (soon after retinal neurogenesis normally initiates).

c. Related to the neurogenesis question, the optic nerve is absent in the panel in Figure 1f. Is this because it is not yet apparent (consistent with delay) or is it on another section? If the latter, then the panel should be replaced so as not to give the impression that its present in the wild type at 48hpf but not in the vsxKO.

7. Related to Point 3 above:

a. A more detailed analysis of the RNA seq data would begin to address this and a good reference for nonretinal gene expression is provided in Rowan et al. (2004). Evidence of ectopic gene expression should be followed by qPCR and in situ hybridization or immunohistochemistry. A later timepoint should also be used, for example at the start of retinal neurogenesis.

8. Related to Point 5 above:

a. This can be addressed in several ways including more markers, birthdating, morphological analysis by EM.

b. It seems that the enhanced apoptosis in the INL at the 72 and 96 hpf timepoints could be indicating that bipolar fated precursors are dying. An early bipolar marker costained with aC3 could reveal this.

c. This is perhaps a long shot but could the apoptosis at 72 and 96 hpf be related to the increased proliferation at 60 and 72 hpf? A pulse of Brdu at ~60 hpf followed by staining at 72 hpf for BrdU and aC3 and showing colocalization could link them. And if this is when bipolar cells are being born, an interesting picture begins to emerge, especially if the three processes can be tied together.

9. The characterization of some cell types was rather cursory in detail. More cell type markers should be presented.

10. The observations made for the ptf1a and prdm1a were not in agreement with the images shown.

11. Related to the RNA seq and ATAC seq data:

a. The choice of 18 hpf for these analyses seems quite early. At what stage is vsx2 expression activated?

b. The decision to use heads for the RNA seq and ATAC seq data could result in a significant loss of resolution for retinal gene expression and chromatin accessibility. A better approach would have been to use a later timepoint when retinas could be dissected away from the RPE and other tissues. Several studies for vsx2 in zebrafish have reported gene expression at 24 hpf. A similar issue exists for the qPCR. Many of the genes analyzed are not necessarily retina specific and changes in expression could be obscured by the presence of other tissues in addition to the very early stage that was used.

c. A table of the differential gene expression analysis from the RNA-seq and a table for the DOCR analysis from the ATAC seq should be provided, preferably the whole datasets.

d. A table showing the 119 genes that are predicted to overlap by DEG and DOCR analysis should be provided.

e. What are the criteria for associating a DOCR with a DEG?

f. Gene enrichment analyses (KEGG, GO, etc) should be done on these different gene cohorts.

12. There are a few instances of omitted or inaccurate reporting of previous findings.

a. The delay in photoreceptor gene expression was described here as a novel finding, but Rutherford et al. (DOI: 10.1167/iovs.03-0332) reported on a similar phenotype in the vsx2 mutant mouse.

b. Lines 444-447: the statement that 'Our observations are also in contrast to previous reports in zebrafish using morpholinos against vsx2, which show microphthalmia and ocular malformations (Barabino et al., 1997; Gago-Rodrigues et al., 2015; Vitorino et al., 2009) is not accurate and incomplete. Barabino et al. used antisense oligonucleotides, not morpholinos, and Clark et al. (PMCID: PMC3315787) was not cited here even though vsx2 and vsx1 morpholinos were used in that study.

c. The vsx1 gene was first reported in 1994, not in 1997 as suggested (https://doi.org/10.1002/cne.903480409).

13. The lethality phenotype of the vsxKO fish is not consistent with any other study on vsx mutants. If the authors can identify the genetic cause, it could be quite informative. Have different genetic backgrounds been tested? The severity of the Vsx2 mutant phenotypes in mice are background dependent. Perhaps this could also explain the lack of microphthalmia in vsxKO fish.

[Editors’ note: further revisions were suggested prior to acceptance, as described below.]

Thank you for resubmitting your work entitled "Mutation of *vsx* genes in zebrafish highlights the robustness of the retinal specification network" for further consideration by *eLife*. Your revised article has been evaluated by Didier Stainier (Senior Editor) and a Reviewing Editor.

The manuscript has been improved but there are some remaining issues that need to be addressed, as outlined below:

Essential revisions:

1) All reviewers were in consensus that additional experiments were not necessary. Reviewers 1 and 2 listed several points that seek clarification, especially on new and supplemental datasets. These points should be addressed in the revised manuscript and in a brief response to the reviewers.

2) The broad questions asked by reviewer 2 will be left to the authors' discretion to include in the revised manuscript.

*Reviewer #1 (Recommendations for the authors):*

The authors have addressed my prior concerns and the inclusion of the additional data has strengthened the study considerably. The study succeeds in revealing important distinctions in early vertebrate retinal development that raises interesting questions about how the retinal GRN functions as a network in different species. While the evidence is now strong for the differences between the phenotypes generated by the Crispr mutants compared to the morpholinos, it still is not clear why this is happening. At this point, however, I believe this issue goes beyond the scope of the present study, especially since the authors provided important controls and data. My comments below are primarily about data accessibility and clarifications of the supplemental datasets.

I will leave this to the authors' discretion, but some of the data in the supplemental figures could fit into the main figures which could help the manuscript flow better. Figure S4 – S6, S9, and S10 come to mind.

I'm not familiar with the PCA plot in Figure S10b. It appears to be a combination of a MA plot and a PCA plot and it's unclear what the vectors for the genotypes are originating from. Some clarification of this type of plot would help, especially given the importance of the point being made in this figure.

The supplemental datasets are hard to follow since they lack titles and descriptions. Detailed descriptions could be provided that are comparable to a figure legend. It could also help if the first worksheet for each dataset file has a key with notes.

With respect to datasets 3 and 4:

1. Dataset 3 appears to be the intersections of differentially expressed genes in the vsxKO with the ATAC data. Is the data referring to gene expression?

2. Dataset 4 appears to contain differential gene expression analysis between the morphant and wild type (worksheet 1) and mutant and morphant (worksheet 2). Is the wild type in worksheet 1 uninjected or control MO-injected?

3. Related to this, were all libraries prepared and sequenced together? If not, direct comparisons can be fraught with batch effect issues that are not easily corrected. If prepared and sequenced at different times, provide documentation and evidence for successful batch effect correction.

4. On this same point, there are less direct, qualitative ways to compare differential gene expression between sample groups that avoid issues with batch effect corrections such as Venn diagrams or other intersectional analyses. This might be sufficient for the point being made if the samples are from different experiments.

5. What is the significance of filtering dataset 4 with V1+V2 values greater than 10?

6. The p and q values in both analyses in dataset 4 are highly repetitive within their respective columns. I'm not used to seeing this, so if it is normal and doesn't need fixing, please provide a statistical explanation for this.

It would be useful if the authors could provide the complete results of the differential expression analysis for vsxKO compared to its respective control condition (wild type or one of the partial combinatorial mutants).

In lines 465-474, the discussion highlighting the differences in Muller glia in the vsxKO compared to the mouse orJ mutant presents reasonable speculation as to the ultimate fate of the missing bipolar cells. A point for the authors to consider is that Rowan et al. (doi: 10.1016/j.ydbio.2004.03.039) noted that their Chx10 BAC reporter was primarily expressed in Muller glia when crossed into the orJ mutant. While they didn't quantify the proportion of Muller glia in the mutant, they did suggest that Vsx2 was preferentially expressed in Muller glia rather than rods.

In Figure 4 legend towards the end, the term "double mutant samples" is used. In keeping with the naming established earlier in the paper, it would be more consistent to refer to them as "vsxKO samples"

*Reviewer #2 (Recommendations for the authors):*

In the revised version of the manuscript "Mutation of Vsx genes in zebrafish highlights the robustness of the retinal specification network" the authors included new time points of their descriptive data, provided protein analysis showing loss of vsx1/vsx2 protein in the mutants and provided RNAseq analysis comparing the transcriptional differences associated with the inconsistent phenotypes between previously published morphants and the newly generated mutant lines. The morphant data helps solidify the differences in early phenotypes and is applicable to all researchers using zebrafish as a model system by further highlighting the importance to be vigilant of the zebrafish's ability to compensate for gene redundancy. Together the new data has strengthened the authors' conclusions and nicely addressed many of the previous reviewer's questions.

---

## [Author Response]

[Editors’ note: the authors resubmitted a revised version of the paper for consideration. What follows is the authors’ response to the first round of review.]

Reviewer #1 (Recommendations for the authors):This is a solid paper that makes a contribution to understanding the roles of Vsx genes in retinal development in vertebrates. There are often 2 or more Vsx genes in different vertebrates so the effects of knocking them out singly can only partially reveal the function of a full Vsx knockout. This manuscript shows that a CRISPR KO of the both Vsx genes in zebrafish leads to retinal phenotypes that are similar in several ways to that of Vsx2 mutant (ocular retardation or) mice, i.e. lack of bipolar cells accompanied fate switching, altered proliferation, and increased cell death.There is also a significant difference between VsxKO zebrafish and or mutant mice. The KO fish do not appear to show microphthalmia while or mutant mice do. To begin to explore this, the authors looked at how Vsx2KO affects other genes in involved in retinal development in zebrafish using sequence analyses and gene ontologies, and they found that although the VsxKO impinges on the regulation of thousands of genes, very few actually showed significant changes at the transcriptome level. Most dysregulated, as would have been expected, were the mutant transcripts coming off the CRISPRed Vsx genes themselves. Although there were also changes in the level of Rx (as has previously reported in Vsx2 MO fish). Interestingly Rx mutants and Vsx2MO fish show microphthalmia.It would have been interesting to compare how the GRN in zebrafish in affected by Vsx2KO to that of the or mutant mice, or even Vsx2 morphants. A difference might reveal why the one shows microphthalmia and the other not.

We thank this reviewer for the positive comments on our work. Regarding the issue of the phenotypic discrepancies observed between *vsx2* morphants and *vsx*KO mutants, in the revised version we are now including the requested RNA-seq data in the new Supplementary Figure S10. (see note 1)

We have confirmed our previous *vsx2* morpholino injection results (see Gago-Rodrigues et al., 2015 Nat Comm, with additional controls; see Supplementary Figure S10), retrieving the microphthalmic phenotypes described by us and other groups (Vitorino et al., 2009; Clark et al., 2008; Barabino et al., 1997). We then performed a comparative RNA-seq analysis of the transcriptional changes in mutants and morphants. Surprisingly, the results clearly showed that, in contrast to the very mild transcriptional changes observed in the double mutants, the neural retina specification network appears strongly upregulated in the morphants; which also display a general downregulation of RPE markers. These results suggest that the dysregulation of the retinal network induced by the morpholinos (likely through compensatory mechanisms that operate at the RNA level) are the molecular cause of the early microphthalmia observed in the *vsx2* morphants.

Unfortunately, we did not examine the expression levels of NR and RPE specifiers in the *vsx2* morphants in our previous work (Gago-Rodrigues et al. 2015 Nat comm). We assumed a downregulation of the neural retina GRN, thus misinterpreting the molecular cause of the phenotype observed in the morphants. Previously, Vitorino et al. had performed expression analyses (by RT-PCR) in *vsx2* morphants for a few NR and RPE markers, such as *foxn4*, *mitf*, *rx3* or *crx,* (Vitorino et al., 2009). However, these analyses are less informative, as were carried out from 55 hpf on, long after the specification of the neural retina and RPE domains, which occurs in the 15-18 hpf window (Buono et al., 2021 Nat comm PMID: 34162866). (see note 2)

Note1: We already had these data before submitting our work to *eLife*. At the time, it was unclear to us to which extent adding these data to the manuscript, which did not alter any of the main conclusions of our work, may deviate the attention from our main findings setting the focus on previous work rather than in the current findings. In retrospect, in the light of the referees’ comments, we realized that not including these experiments in the initial submission was a mistake.

Note 2: I would like to emphasize that by any means our observations attempt to disregard previous *vsx2* morpholino studies (among others by us) that consistently reported microphthalmia in zebrafish morphants. These were well controlled and reproducible studies in which the molecular causes of the observed eye-specific phenotype were simply misinterpreted (likely due to the phenotypic descriptions in mutant mice).

Is it possible to get transcriptomic data from or mutant mice or Vsx2 morphant fish for comparison of the genetic robustness and clues to the or phenotype and why it isn't seen in Vsx2KO?

We are including the *vsx2* morphant RNA-seq data requested by the referee in the new Supplementary Figure S10.

I would have liked to have seen the INL in better detail, for example using Vsx reporter lines. So one could see what happens to the fate switched bipolar cells.

We agree in that following the precursors by live imaging would have help to visualize the differentiation trajectories. Unfortunately, the time required to set the crosses of reporter lines in the double mutant background to obtain homozygous *vsx*KO goes beyond the re-submission window. Nevertheless, we have added new molecular markers to our analyses and examined additional developmental stages to strengthen our conclusions on the fate of the precursors failing to differentiate as bipolar cells (see Supplementary Figures S6 and Figure S7).

Reviewer #2 (Recommendations for the authors):In Letelier et al. the authors investigate the functional conservation of the visual system homeobox transcription factors vsx1 and vsx2 using zebrafish as a model for the teleost species. Genetic mutations were generated for each gene through CRISPR-Cas9 targeting of the conserved DNA binding domain, which resulted in a vsx1 frameshift mutation and vsx2 early stop codon. Individual vsx1 or vsx2 homozygous mutants are viable and only vsx1 -/- animals present ocular phenotypes, specifically a decrease in visual function based on electroretinograms, the visual background adaptation reflex, and optokinetic response. The lack of phenotypes, such as small eyes, in vsx2-/- animals is a departure from mammalian models of vsx2 loss of function. The vsx1-/-;vsx2-/- animals also do not feature small eyes but do present other retinal developmental phenotypes. These include increased outer nuclear layer thickness, decreased inner nuclear layer thickness, visual defects, increased cell death, prolonged proliferation, delayed photoreceptor differentiation, and changes to various cell fates. Finally, ATACseq analysis of open chromatin in vsx1-/-;vsx2-/- animals revealed significant changes to open chromatin while RNAseq only resulted in minor changes to RNA transcripts. Based on the collection of results, the authors conclude early zebrafish visual system development differs from mammals due to compensatory mechanisms built into the eye field gene regulatory network.The strengths of this manuscript include the high quality data presented and the variety of techniques used to investigate the changes to vision and retinal histogenesis in vsx1;vsx2 double homozygous mutants. Despite these strengths there are other mechanistic explanations for the differences between zebrafish and mammalian vsx1;vsx2 function, which can be addressed with additional analysis.

We have carefully considered all the comments raised and we have made an effort to address all of them in detail.

1. Due to the lethality of vsx1-/-;vsx2-/- animals, the only way to produce double homozygous offspring is to use parents with a wild type allele of either gene. This leads to the potential of maternal contributions of the wild type allele during early development, including the time window of optic cup formation. A maternal contribution of either vsx1 or vsx2 could contribute to the lack of early phenotypes, such as small eyes. The lack of microphthalmia in the mutants studied may be due to the maternal contribution of functional vsx1 or vsx2 genes depending on the breeding scheme.

A potential maternal contribution by *vsx* genes (i.e., that could rescue the early eye phenotype) is a very unlikely hypothesis. Despite a previous report claiming *vsx* genes having such a maternal contribution (Xu et al., Dev Biol 2014), recent data from several independent groups using RNA-seq profiling throughout early zebrafish development clearly demonstrates that *vsx1* and *vsx2* are not maternally contributed**.** See information from the Zebrafish Expression atlas (https://www.ebi.ac.uk/gxa/home) Author response image 1: White et al., 2017, *eLife* PMID: 29144233. that was also confirmed in the datasets provided in Vejnar et al., 2019, PMID: 31227602, Genome Res; and Marletaz et al., 2018 Nature; (PMID: 30464347). Furthermore, we could confirm by western blot that Vsx1 is not maternally contributed as a protein (Supplementary Figure S1). Finally, the fact that double mutant embryos were obtained by in-crossing of either *vsx1∆245+/-; vsx2∆73-/-* or *vsx1∆245-/-; vsx2∆73+/-* animals, rules out any possibility for a maternal rescue of Vsx function.

**Author response image 1. sa2fig1:** 

2. The vsx1 mutation is an in frame deletion and despite the loss of the DNA binding domain, several other regions of conserved sequence are retained. In addition, vsx1 is one of three genes significantly upregulated in the RNAseq analysis with a log2 fold change > |1.5|in vsx1-/-;vsx2-/- animals. The qPCR analysis of eye related genes resulted in a 7 fold increase in vsx1. This large increase in the mutant form of vsx1 may be masking phenotypes or inducing different phenotypes than true loss of function mutations.

The hypothesis of *vsx1* retaining some function after the deletion of critical residues of the DNA binding domain and part of the CVC domain (see Supplementary Figure S1) seems also quite unlikely, as point mutations compromising the homeodomain DNA binding affinity have been proved essential for Vsx function. Mutation of critical Arginine residues in the Vsx2 homeodomain behave phenotypically as a null allele in mice (Zou and Levin 2012, PMID: 23028343). Thus, considering the high sequence conservation at the homeodomain in Vsx proteins, and the large deletion generated in the *vsx1* allele here described it is logic to assume that our allele behaves also as a null.

Additionally, although *vsx1* mRNA is upregulated in *vsx*KO mutants, the levels of the predicted in-frame Vsx1 truncated protein are undetectable, as we have determined by western blot (Supplementary Figure S1). This, together with the recessive nature of the *vsx1* mutation, argue against any kind of neomorphic effect in our Vsx double mutants.

3. The authors show the ONL of vsx1-/-;vsx2-/- retinas are larger compared to controls, yet there are no clear changes to the photoreceptor precursor marker prdm1a (Figure 4). Further, even though there is a delay in photoreceptor differentiation there are no visible differences in expression of Zpr1+ cones and Zpr3+ rods (Figure S4). It is unclear based on these set of results to what is causing the increased size of the ONL in vsx1-/-;vsx2-/- retinas.

We agree with the referee’s comment. In the previous version, we quantify the increased thickness of the ONL at 6 dpf, whereas Zpr1 and Zpr3 stainings were examined only at 72 and 96 hpf. To address this point, we extended our observations to quantify Zpr1 and Zpr3 staining also at 6 dpf. The new data, showing a significant increase of photoreceptors’ associated staining in *vsx*KO retinas, are now included in Supplementary Figure S6.

4. The authors should address the issue of maternal contribution of wild type vsx1 and vsx2. This can be difficult to do experimentally, especially due to the mutations not completely eliminating vsx1 and vsx2 through nonsense mediated decay. If it is not possible to assess maternal wild type mRNA or protein the authors should comment on this issue within the manuscript text.

As mentioned before, a potential maternal contribution by *vsx* genes is a very unlikely hypothesis as RNA-seq data convincingly show these genes are not maternally provided. Please see full comments above. We have included the following sentence in the discussion to refer to this possibility. “…retina identity in zebrafish. Although a potential rescue by maternally provided *vsx* genes could be hypothesized as an explanation for normal specification of the retina, this is an unlikely possibility as both transcripts are not detectable before zygotic genome activation (White et al., 2017, *eLife* PMID: 29144233”).

5. In relation to point 1, while there are minimal changes to the transcriptome at 18/19hpf there are changes to bipolar and MG cells, suggesting late transcriptional changes in the mutants. Does this provide further evidence for maternal contribution? Why does genetic redundancy buffer the early but not late phenotypes? Is otx2a or otx2b expression changed in the mutants?

A fundamental difference between the NR specification GRN and the bipolar cells network is that, during the early specification of the tissue, numerous homeodomain TFs (i.e., including not only *vsx1* and *vsx2*, but also *rx1*, *rx2*, *rx3*, *lhx2b*, *lhx9*, *hmx1*, and *hmx4*) converge on the same *cis*-regulatory modules through the 5’-TAATT-3’ motif, increasing the robustness of the network (Buono et al., 2021, Nat comm PMID: 34162866). We already commented on this important aspect in the Discussion section. Regulatory redundancy is even more pronounced in teleost species, due to the existence of additional paralogs after the extra round of genome duplication specific of this clade. The scenario is quite different for the bipolar cells’ specification network, as many of these factors are no longer expressed in precursors committed to the bipolar lineage but acquire specialized functions in the differentiation of other neuronal types. This segregation of the eye specifiers into cell-specific networks, makes bipolar specification critically dependent on *vsx* genes’ function.

Answering the second question: according to our RNA-seq data set at stage 18hpf, the expression of *otx* genes is not significantly different in *vsx*KO mutants when compared to wild type animals (See supplementary dataset 3).

5. Overexpression experiments with the mutant form of vsx1 should be performed to address whether or not it produces a retinal phenotype similar to what is observed in the mutants. A large presence of the Vsx1 mutant protein may produce ectopic interactions otherwise not observed.

As already mentioned, it is very unlikely that *vsx1* retains some function after the deletion of critical residues of its DNA binding domain and part of the CVC domain (see Supplementary Figure S1), for point mutations in critical Arginines are sufficient to block Vsx function (Zou and Levin 2012, PMID: 23028343). On the other hand, we find challenging the interpretation of an overexpression experiment as suggested by the referee (e.g., by injecting mRNA at one-cell stage). It is important to consider that *vsx*KO mutants do not display an early phenotype in retinal specification, and that the late phenotypes on bipolar differentiation are not apparent before 72 hpf, when the injected mRNA would no longer be present in the embryo. In addition, although *vsx1* expression is elevated in the mutants, the new Western blot data show similar if not lower Vsx1 protein levels (Supplementary Figure S1). These considerations, together with the fact that toxicity and artifacts are also a possibility in overexpression experiments, discourage us to attempt the suggested experiments.

6. Another interesting observation from the study is the increase in MG cells based on GFAP staining. A previous study from Hatakeyama et al. (2001 Development 128 (8): 1313-1322.) found misexpression of Vsx2 (Chx10) in explant cultures increased the numbers of MG cells. Livine-Bar et al. (2006. PNAS 103 (13) 4988-4993) performed a similar experiment and although they didn't report a significant increase in MG cells, the bar graph containing cell counts does show a trend towards increased MG (Figure 1D).

We thank this reviewer for pointing to us the article by Hatakeyama et al., 2001, which further supports the proximity of the Müller glia and bipolar cells differentiation trajectory. We have now included this reference in the discussion and changed the corresponding paragraph accordingly to also mention the observed trend:

“Both glial and bipolars cells are late-born retinal types deriving from a common pool of precursors with restricted developmental potential (Bassett and Wallace, 2012; Satow et al., 2001; Hatakeyama et al. 2001). In mice, however, a significant increase in Müller glia cells has not been reported in experiments genetically interfering with *vsx* either postnatally (Goodson et al., 2020; Livne-Bar et al., 2006) or specifically in bipolar cells (Norrie et al., 2019)”

“Alternatively, the increase may have been overlooked in previous studies due to the small size of the Müller glia cell population. The fact that a trend towards an increase in Müller glia has been reported (Livne-Bar et al., 2006) may support this second possibility”.

Is there a role for the truncated version of the mutant Vsx2 generated in this study? While vsx2 levels are decreased via qPCR in the double mutants, the in situ in Figure S8 shows a significant increase in vsx2 expression at 72 hpf.

It is extremely unlikely that the truncated Vsx2 protein is functional. As we argue previously for *vsx1*, the transcriptional properties of members of this family crucially depend on the homeodomain, a domain that is severely truncated in the mutants. In addition, mRNA levels are significantly reduced at 18hpf, and no protein could be detected by western blot in *vsx*KO mutants at 24hpf (Supplementary Figure S1). Furthermore, in our *vsx2* mutant the CRISPR deletion comprised critical conserved Arginines (R200 and R227) causative of microphthalmia in mouse and human patients (see Supplementary Figure S1).

The fact that *vsx2* mRNA levels appear upregulated at 72 hpf would argue for a dynamic regulatory logic during development. We have already commented on this in the following paragraph of the discussion: “Vsx biphasic activity follows a partially independent *cis*-regulatory control by enhancers active either in precursors, bipolar cells, or both (D. S. Kim et al., 2008; Norrie et al., 2019; Rowan and Cepko, 2005)”.

7. Cell counts of rods and cones will help clarify the disparity in staining abundance and ONL thickness changes. There is precedent for an increase in rods with late knockdown of Vsx2 expression (Livine-Bar et al. 2006).

We agree with the referee’s comment. We have addressed this point by quantifying Zpr1 and Zpr3 staining at 6 dpf, when the ONL was significantly wider. The new data, showing a significant increase of photoreceptors’ associated staining in *vsx*KO mutant retinas, are now included in Supplementary Figure S6.

Reviewer #3 (Recommendations for the authors):Letelier and colleagues examined the genetic requirements of the two paralogous VSX genes, vsx1 and 2, in zebrafish retinal development by generating Crispr deletions that target each gene. A single copy of either gene is sufficient to prevent macroscopic changes or lethality, but double mutants (vsxKO) die at approximately two weeks. Most of the study then centers on the retinal phenotypes of the single and vsxKO phenotypes, largely focusing on the known phenotypic characteristics of vsx1 and vsx2 mutants or knockdowns in other species such as mouse, medaka, and chick. Notable differences in zebrafish versus the other species were the lack of microphthalmia, retinal hypocellularity in the early retina, and ectopic pigmentation. These phenotypes are typically due to vsx2 loss of function, and the authors show that genetic compensation or redundancy between vsx1 and vsx2 is not the reason. A similar lack of compensation/redundancy between vsx1 and vsx2 for these early retinal phenotypes was previously shown in mouse and was suggested in zebrafish. Notable similarities were the reduction in bipolar cell function and the resulting defects in visual signaling as noted by ERG and the visual background adaptation (VBA) reflex. Other cross-species similarities were delayed differentiation of photoreceptors, and a skew in the proportions of late cell types, notably rods and Muller glia. Bulk RNA sequencing and ATAC seq in 18 hpf wild type and mutant retina demonstrated a limited degree of overlap with respect to differentially expressed genes and nearby differentially accessible chromatin regions. These last findings combined with the lack of microphthalmia and apparent lack of changes in early retinal development led the authors to suggest that these differences highlight the robustness of the retinal specification network.With a couple of exceptions, the phenotypic characterizations and interpretations are supported with data, but the depth of analysis is generally not sufficient to reveal mechanistic insights or push the field beyond what has already been characterized.

We thank this reviewer for the positive comments on our work. We have considered in detail all the comments and suggestions, and we feel confident that have address most if not all of them.

The main conclusion that the work provides insight into the robustness of the retinal specification network is not supported by data. The lack of an early retinal phenotype reported here is not consistent with other reports of vsx knockdown in zebrafish. The speculation that the differences are due to morpholinos versus genetic manipulation is concerning because no proof of this is provided. In that this outcome calls into question prior research by others and one of the senior authors of this study, it is imperative to provide an explanation supported by data.

We thank this reviewer for the positive comments on our work. Regarding the issue of the phenotypic discrepancies observed between *vsx2* morphants and *vsx*KO mutants, in the revised version we are now including the requested RNA-seq data in the new Supplementary Figure S10. (see note 1)

We have confirmed our previous *vsx2* morpholino injection results (see Gago-Rodrigues et al., 2015 Nat Comm, with additional controls; see Supplementary Figure S10), retrieving the microphthalmic phenotypes described by us and other groups (Vitorino et al., 2009; Clark et al., 2008; Barabino et al., 1997). We then performed a comparative RNA-seq analysis of the transcriptional changes in mutants and morphants. Surprisingly, the results clearly showed that, in contrast to the very mild transcriptional changes observed in the double mutants, the neural retina specification network appears strongly upregulated in the morphants; which also display a general downregulation of RPE markers. These results suggest that the dysregulation of the retinal network induced by the morpholinos (likely through compensatory mechanisms that operate at the RNA level) are the molecular cause of the early microphthalmia observed in the *vsx2* morphants.

Unfortunately, we did not examine the expression levels of NR and RPE specifiers in the *vsx2* morphants in our previous work (Gago-Rodrigues et al. 2015 Nat comm). We assumed a downregulation of the neural retina GRN, thus misinterpreting the molecular cause of the phenotype observed in the morphants. Previously, Vitorino et al. had performed expression analyses (by RT-PCR) in *vsx2* morphants for a few NR and RPE markers, such as *foxn4*, *mitf*, *rx3* or *crx,* (Vitorino et al., 2009). However, these analyses are less informative, as were carried out from 55 hpf on, long after the specification of the neural retina and RPE domains, which occurs in the 15-18 hpf window (Buono et al., 2021 Nat comm PMID: 34162866). (see note 2)

Note1: We already had these data before submitting our work to *eLife*. At the time, it was unclear to us to which extent adding these data to the manuscript, which did not alter any of the main conclusions of our work, may deviate the attention from our main findings setting the focus on previous work rather than in the current findings. In retrospect, in the light of the referees’ comments, we realized that not including these experiments in the initial submission was a mistake.

Note 2: I would like to emphasize that by any means our observations attempt to disregard previous *vsx2* morpholino studies (among others by us) that consistently reported microphthalmia in zebrafish morphants. These were well controlled and reproducible studies in which the molecular causes of the observed eye-specific phenotype were simply misinterpreted (likely due to the phenotypic descriptions in mutant mice).

1. The generation of the Crispr deletions is an important step for better understanding the genetic requirements of vsx1 and vsx2 in zebrafish. More information is needed, however, to understand the true nature of the mutations, This should include a graphic showing how the mutant alleles are altered with respect to open reading frames, domains, and structural motifs. Data should also be provided demonstrating the degree of mutant protein expression, preferably by western blot. This could then allow the authors to determine whether these alleles are nulls, hypomorphs, neomorphs, etc. This issue could also be relevant to understanding why the vsx2 mutant differs from morpholino knockdown

We do agree with this referee in that including more information on the domains affected by the deletions will improve the clarity of the work. A new scheme has been generated and is now shown in Supplementary Figure S1.

In the new Supplementary Figure S1 we also show protein expression by western blot using Vsx1 and Vsx2 specific antibodies. We show that mutant Vsx2 and Vsx1 proteins cannot be detected by WB, suggesting reduced stability. As already pointed out to referee 2, it is very unlikely that this mutant Vsx1 protein retains some function: particularly after the deletion of its DNA binding domain and part of the CVC domain (see Supplementary Figure S1). It has been described that the mutation of critical Arginines in the homeodomain is sufficient to block Vsx function leading to a null allele (Zou and Levin 2012, PMID: 23028343). In our *vsx2* mutant, the deletion comprised those conserved critical amino acids causative of microphthalmia in mouse and human patients (see Supplementary Figure S1).

2. The lack of microphthalmia is an unexpected outcome. The apparent lack of a reduced proliferation phenotype at 48 hpf and a lack of overt pigmentation in the retina suggest that the early retinal phenotypes observed in vsx2 mutant mice, medaka, and vsx2(R200Q) mutant human organoids are not occurring in zebrafish. But the earliest stages of retinal development were not presented. It is well established that the proliferation and identity defects in vsx2 mutant mice are revealed very soon after vsx2 onset. In addition, the initiation of neurogenesis is delayed in mouse vsx2 mutants. If the authors performed an earlier phenotypic analysis and their interpretation holds, this would be one of the more novel aspects of the study. And the study would be greatly strengthened by data that provide new mechanistic insights and/or future directions for the field.

In the revised version of the manuscript, we have followed all the reviewer suggestions and extended our observations to early developmental stages. All the data have been included in the new Supplementary Figure S5. We show that, in agreement with our previous observations at 48hpf, no significant differences in proliferation rates are detected at 24 hpf, as examined by PH3 staining. Interestingly, we found that the onset of *atoh7* expression is slightly but consistently delayed in the double mutants; as it has been reported in mice. These data, together with the correct specification of the RPE domain (see point below and Supplementary Figure S5), support our hypothesis of a normal specification of the optic cup domains in *vsx*KO mutants. Thus, this information has now been included in the manuscript (Supplementary Figure S5). We thank the reviewer for pointing us in this direction.

3. A lack of pigmentation does not rule out changes in gene expression that would indicate problems with retinal identity. Pigmentation is the outcome of a differentiation pathway, whereas identity issues could be indicated by the ectopic expression of genes related to other tissues such as RPE, ciliary epithelium, etc.

This aspect was already addressed by our comparative RNAseq analysis of WT and *vsx*KO mutant embryos at 18 hpf. None of the RPE specifiers were significantly dysregulated in the *vsx*KO embryos during the specification of the optic cup domains (See supplementary dataset 3). To further confirm this, we examined the expression of two core components of the RPE GRN: *bhlhe40* and *tfec* by fluorescent ISH. This analysis (now included in Supplementary Figure S5) shows normal expression of both markers in *vsx*KO mutants.

4. Issues 2 and 3 pertain to the early retinal phenotypes observed with multiple alleles in mice, medaka and to a certain extent with results in zebrafish vsx2/chx10 morphants. The apparent differences across species could very well be interesting, but the differences between prior morpholino data versus the mutant data here is troubling. First, Clark et al. (PMCID: PMC3315787) showed obvious changes in cyclind1 and vsx1 in chx10 morphants at 24 hpf, a finding very consistent across species. Second, this is concerning in its implication for one of the senior author's prior work. Gago-Rodrigues et al. (DOI: 10.1038/ncomms8054) performed well controlled experiments showing that their vsx2 knockdown was specific and used this approach to demonstrate microphthalmia and defective optic cup morphogenesis through a mechanism positing that vsx2 promotes the expression of ojoplano (opo). It was a rigorous study and very consistent with what the corresponding author's group reported in medaka. The lack of data to support a suitable explanation diminishes the findings presented here. If the authors believe it is due to redundancy by a non-vsx gene, then that should be shown.

We have addressed this point in detail in the reply to the general comments of this referee (see above), as well as in those to referee #1. In the revised version we are now including RNA-seq data also comparing the transcriptome of wild type and *vsx2* morphants at 18 hpf. These data, which explain the phenotypic discrepancies between mutants and morphants, have been included in the Supplementary Figure S10.

5. The data for visual impairment and reduced bipolar function is strong. However, evidence of the fate of the bipolar cells is lacking. In germline vsx2 mouse mutants and early postnatal Crispr inactivations, the bipolar cells fail to form or rapidly die. In late postnatal Crispr inactivation and in vsx1 mutants, bipolar cells form but don't differentiate properly and their function is impaired. More direct evidence of how bipolar cell formation or function is affected should be provided.

Our data point to multiple fate alternatives for precursors that fail to differentiate as bipolar cells: such as undergoing apoptosis, extending their proliferative phase, or differentiating, as photoreceptors or Müller glia cells. Therefore, to address this issue properly would have required following individual precursors by live imaging to visualize their differentiation trajectories, which is beyond the initial objectives of this work.

A possibility could have been crossing the mutant line with different reporters and performing imaging analyses. Unfortunately, the time required to set the crosses in the double mutant background to obtain homozygous *vsx*KO mutants goes over the re-submission window. Nevertheless, following the reviewer comments, we have added new molecular markers to our analyses and examined additional developmental stages to strengthen our general conclusions on the fate of the precursors failing to differentiate as bipolar cells (see Supplementary Figures S6 and S7).

6. Related to Point 2 above:a. At the minimum, a 24 hpf timepoint should be analyzed with proliferation and cell cycle markers (e.g. pHH3, cyclind1 BrdU, pcna) and expression for RPE/pigmentation genes (e.g. mitf, otx1, dct, to name a few) should be done.b. The delay of neurogenesis in mouse vsx2 mutants is clearly indicated by markers of retinal ganglion cells, Tuj1 staining, and markers neurogenic progenitors (e.g. atoh7, neurog2, ascl1, otx2 (also marks photoreceptor precursors)). These or equivalent zebrafish markers should be assessed at the appropriate timepoint (soon after retinal neurogenesis normally initiates).c. Related to the neurogenesis question, the optic nerve is absent in the panel in Figure 1f. Is this because it is not yet apparent (consistent with delay) or is it on another section? If the latter, then the panel should be replaced so as not to give the impression that its present in the wild type at 48hpf but not in the vsxKO.

All these aspects have been addressed in the revised version as discussed above. PH3 stainings, and ISH have been performed to examine proliferation, differentiation and RPE specification in the mutants. The information is now summarized in Supplementary Figure S5. Regarding the point 1c, we have substituted the panel in Figure 1f. The optic nerve develops normally in the *vsx*KO mutants, as clearly shown in Supplementary movie 1.

7. Related to Point 3 above:a. A more detailed analysis of the RNA seq data would begin to address this and a good reference for nonretinal gene expression is provided in Rowan et al. (2004). Evidence of ectopic gene expression should be followed by qPCR and in situ hybridization or immunohistochemistry. A later timepoint should also be used, for example at the start of retinal neurogenesis.

As stated in point 3, our comparative RNAseq analysis of WT and *vsx*KO mutant embryos at 18 hpf is addressing this point specifically. We apologize for unintentionally did not upload Supplementary data sets 1, 2 and 3 with the first version of the manuscript. This important dataset will be now uploaded (together with a new Supplementary dataset 4 showing the *vsx2* morphants transcriptomics). As pointed out, none of the RPE specifiers were significantly dysregulated in the *vsx*KO embryos during the specification of the optic cup domains. We have further confirmed this point by examining the expression of two core components of the RPE GRN: *bhlhe40* and *tfec* by ISH. This analysis (now included in Supplementary Figure S5) shows normal expression of both markers in *vsx*KO mutants.

8. Related to Point 5 above:a. This can be addressed in several ways including more markers, birthdating, morphological analysis by EM.b. It seems that the enhanced apoptosis in the INL at the 72 and 96 hpf timepoints could be indicating that bipolar fated precursors are dying. An early bipolar marker costained with aC3 could reveal this.c. This is perhaps a long shot but could the apoptosis at 72 and 96 hpf be related to the increased proliferation at 60 and 72 hpf? A pulse of Brdu at ~60 hpf followed by staining at 72 hpf for BrdU and aC3 and showing colocalization could link them. And if this is when bipolar cells are being born, an interesting picture begins to emerge, especially if the three processes can be tied together.

As we mentioned in the comments to the public review (point 5), our data point to multiple fate alternatives for precursors failing to differentiate as bipolar cells: apoptosis, extended proliferation, and alternative fate acquisition as photoreceptors or Müller glia cells. We think that to ultimately address the differentiation trajectories in the mutants would have required following individual precursors by live imaging; something that is out of the scope of our work; which already covers many aspects using a broad methodological approach. Nevertheless, to strengthen our general conclusions on the fate of the precursors we have added now a few more molecular markers to our analyses (*pax6* and *prox1*) and examined additional developmental stages for Zpr1 and Zpr3 stainings (see Supplementary Figures S6 and S7).

9. The characterization of some cell types was rather cursory in detail. More cell type markers should be presented.

This relates to the previous point. We have added a few more markers to the analysis to strengthen our general conclusions on the fate of the precursors (see Supplementary Figures S6 and S7).

10. The observations made for the ptf1a and prdm1a were not in agreement with the images shown.

We partially disagree with the referee in this specific point. We think that images included in Figures 4 and S7 (Supplementary Figure S5 in the previous version) showing *ptf1a* expression at two different stages clearly indicate similar levels of expression in wild type and *vsx*KO embryos. However, we concede to the referee that *prdm1a* expression levels are slightly downregulated in the mutants (Figure 4). This downregulation will be in agreement with the delayed differentiation of the photoreceptors we observed in the mutants (Supplementary Figure S6). We have now changed the text to acknowledge this *prdm1a* mild downregulation in the mutants.

11. Related to the RNA seq and ATAC seq data:a. The choice of 18 hpf for these analyses seems quite early. At what stage is vsx2 expression activated?b. The decision to use heads for the RNA seq and ATAC seq data could result in a significant loss of resolution for retinal gene expression and chromatin accessibility. A better approach would have been to use a later timepoint when retinas could be dissected away from the RPE and other tissues. Several studies for vsx2 in zebrafish have reported gene expression at 24 hpf. A similar issue exists for the qPCR. Many of the genes analyzed are not necessarily retina specific and changes in expression could be obscured by the presence of other tissues in addition to the very early stage that was used.

In our previous work (Buono et al. 2021, Nat comm PMID: 34162866) we analyzed in detail the specification of the neural retina and RPE networks by examining both transcriptional dynamics and chromatin accessibility in each domain. From our data, as well as from anatomical observations (Li et al., 2000, PMID: 10822269; Kwan et al., 2012, PMID: 22186726), it is easy to conclude that it is precisely within the 15 to 18 hpf window that the two GRNs bifurcate and the cells acquire morphological features specific of each domain. We took all this into consideration to choose 18hpf as the optimal stage for RNAseq and ATACseq studies. The use of heads as starting material was also coherent. First the optic vesicles represent almost two thirds of the head volume at this particular stage. In addition, once the neural tube is excluded, the neural retina is the only domain in which *vsx* genes are expressed at these early stages. Thus, it is logical to assume that will be mostly the retinal tissues those affected by the mutation. Taking all together, we think that the presence of additional tissues in the samples will have a minimal masking effect when it comes to analyze the status of the neural retina and RPE networks: particularly because many of the key specifiers in each compartment are also eye-specific genes.

c. A table of the differential gene expression analysis from the RNA-seq and a table for the DOCR analysis from the ATAC seq should be provided, preferably the whole datasets.d. A table showing the 119 genes that are predicted to overlap by DEG and DOCR analysis should be provided.e. What are the criteria for associating a DOCR with a DEG?f. Gene enrichment analyses (KEGG, GO, etc) should be done on these different gene cohorts.

We apologize to all the reviewers for, during the submission, we forgot uploading the necessary Supplementary dataset 1 (Differentially opened chromatin regions; DORCs), Supplementary dataset 2 (enriched GO terms in genes associated to DOCRs) and Supplementary dataset 3 (Differentially expressed genes, DEGs) as items attached to the manuscript. We are now providing these important datasets, together with a new one (Supplementary dataset 4), to include DEGs in *vsx2* morphants. In the revised version of the manuscript, Supplementary Figure S7 has been removed from the manuscript as the information contained in that figure is redundant with Supplementary dataset 2.

Following the referee’s suggestion, a table with the overlap between DEGs and genes associated to DOCRs has also been added to the Supplementary dataset 3. The criteria for DORCs gene assignment was already included in the methods “All the DOCRS have been associated with genes using the online tool GREAT (McLean et al., 2010) with the option “basal plus extension”. Finally, we perform an analysis of GO enriched terms in the set of 119 overlapping genes. This analysis did not yield GO terms with an adjusted *p* value < 0.05 and thus the data has not been included in the manuscript.

12. There are a few instances of omitted or inaccurate reporting of previous findings.a. The delay in photoreceptor gene expression was described here as a novel finding, but Rutherford et al. (DOI: 10.1167/iovs.03-0332) reported on a similar phenotype in the vsx2 mutant mouse.

We thank the reviewer for calling our attention on this article. We are now including a reference to it in the Discussion section.

b. Lines 444-447: the statement that 'Our observations are also in contrast to previous reports in zebrafish using morpholinos against vsx2, which show microphthalmia and ocular malformations (Barabino et al., 1997; Gago-Rodrigues et al., 2015; Vitorino et al., 2009) is not accurate and incomplete. Barabino et al. used antisense oligonucleotides, not morpholinos, and Clark et al. (PMCID: PMC3315787) was not cited here even though vsx2 and vsx1 morpholinos were used in that study.c. The vsx1 gene was first reported in 1994, not in 1997 as suggested (https://doi.org/10.1002/cne.903480409).

We have now included in the text the missing reference and corrected the corresponding sentences.

13. The lethality phenotype of the vsxKO fish is not consistent with any other study on vsx mutants. If the authors can identify the genetic cause, it could be quite informative. Have different genetic backgrounds been tested? The severity of the Vsx2 mutant phenotypes in mice are background dependent. Perhaps this could also explain the lack of microphthalmia in vsxKO fish.

A majority of the published mutations severely compromising vision in zebrafish are lethal during larval stages and/or require extraordinary measures to be further raised to adulthood. This has been reported even if the mutations affect eye-specific genes, such as *atho7* (Neuhaus et al., 1999, PMID: 10493760), suggesting that lethality is likely linked to a compromised feeding behavior. In our case, we have managed to raise only a single double mutant escaper after numerous attempts. In addition to our previous efforts, for this revision we have raised 90 larvae with a compromised visual background adaptation response (3 tanks with 30 larvae each from independent crosses, 30 larvae raised each week); and after 2 months, none of the survival fish were *vsx* double mutants.

[Editors’ note: what follows is the authors’ response to the second round of review.]

Reviewer #1 (Recommendations for the authors):The authors have addressed my prior concerns and the inclusion of the additional data has strengthened the study considerably. The study succeeds in revealing important distinctions in early vertebrate retinal development that raises interesting questions about how the retinal GRN functions as a network in different species. While the evidence is now strong for the differences between the phenotypes generated by the Crispr mutants compared to the morpholinos, it still is not clear why this is happening. At this point, however, I believe this issue goes beyond the scope of the present study, especially since the authors provided important controls and data. My comments below are primarily about data accessibility and clarifications of the supplemental datasets.

We thank the reviewer for the positive comments on our work. In this version of the manuscript all the comments raised by the reviewer have been addressed.

I will leave this to the authors' discretion, but some of the data in the supplemental figures could fit into the main figures which could help the manuscript flow better. Figure S4 – S6, S9, and S10 come to mind.

As suggested by the reviewer, we were thinking extensively with all the authors to include some of the supplementary figures as main figures in the manuscript. However, as in the *eLife* online version of the article, supplementary figures are linked to main figures as *figure supplements*, we think this format will help substantially the manuscript to flow better.

I'm not familiar with the PCA plot in Figure S10b. It appears to be a combination of a MA plot and a PCA plot and it's unclear what the vectors for the genotypes are originating from. Some clarification of this type of plot would help, especially given the importance of the point being made in this figure.

The PCA plot presented in Figure 5—figure supplement 2b (former Figure S10b) is the default graphical output obtained by the PCA plot function of the R package Cummerbund (Goff L, Trapnell C, Kelley D, 2022. cummeRbund: Analysis, exploration, manipulation, and visualization of Cufflinks high-throughput sequencing data. R package) where the dots represent expression values transformed into PC and the vectors show the direction of the variation depending on the PC. However, we agree with the reviewer that this kind of plot could be misleading, and it does not add much to the understanding of the represented results. Hence, we decided to modify the Figure 5—figure supplement 2b (former Figure S10b) graph and represent the data as a standard PCA plot with no dots or vectors.

The supplemental datasets are hard to follow since they lack titles and descriptions. Detailed descriptions could be provided that are comparable to a figure legend. It could also help if the first worksheet for each dataset file has a key with notes.

As suggested by the reviewer, a detailed description has been added in the first worksheet of each Supplementary Dataset.

With respect to datasets 3 and 4:1. Dataset 3 appears to be the intersections of differentially expressed genes in the vsxKO with the ATAC data. Is the data referring to gene expression?

Yes, the Figure 5-source data 3 (former Supplementary Dataset 3) is the intersection of differentially expressed genes (DEGs) in the *vsx*KO (vs WT) with the significantly differentially open chromatin regions (DOCRs) from ATAC-seq, that refers to gene expression. To complement that dataset, a complete list of DEGs (*vsx*KO vs WT) with relative expression data has been added as a new worksheet in the same Figure 5-source data 3.

2. Dataset 4 appears to contain differential gene expression analysis between the morphant and wild type (worksheet 1) and mutant and morphant (worksheet 2). Is the wild type in worksheet 1 uninjected or control MO-injected?

As stated in the legend of Figure 5—figure supplement 2 (former Figure S10), WT (uninjected) animals were used as controls in this experiment. To further clarify this point, we included that information in Figure 5-source data 4 legend (former Supplementary Dataset 4). We didn´t use control MOs for this experiment as the efficiency of the splicing morpholino was assessed directly in the RNA-seq experiments (see Figure 5—figure supplement 2f).

3. Related to this, were all libraries prepared and sequenced together? If not, direct comparisons can be fraught with batch effect issues that are not easily corrected. If prepared and sequenced at different times, provide documentation and evidence for successful batch effect correction.

We thank the reviewer to raise this important point. As stated in Note1 of the first rebuttal letter (after the initial round of revisions), we had the morpholino data before submitting our work to *eLife*, so no batch effect is observed as all RNA-seq and ATAC-seq libraries were prepared and sequenced together.

These following two paragraphs comes from the initial rebuttal letter:

“We thank this reviewer for the positive comments on our work. Regarding the issue of the phenotypic discrepancies observed between vsx2 morphants and vsxKO mutants, in the revised version we are now including the requested RNA-seq data in the new Supplementary Figure S10. (see note 1)”

Note1: We already had these data before submitting our work to *eLife*. At the time, it was unclear to us to which extent adding these data to the manuscript, which did not alter any of the main conclusions of our work, may deviate the attention from our main findings setting the focus on previous work rather than in the current findings. In retrospect, in the light of the referees’ comments, we realized that not including these experiments in the initial submission was a mistake.

4. On this same point, there are less direct, qualitative ways to compare differential gene expression between sample groups that avoid issues with batch effect corrections such as Venn diagrams or other intersectional analyses. This might be sufficient for the point being made if the samples are from different experiments.

Please refer to the previous reply (point 3). No batch effect is possible as all RNA-seq and ATAC-seq libraries were prepared and sequenced together.

5. What is the significance of filtering dataset 4 with V1+V2 values greater than 10?

Filtering out genes that have all zero expression values or very low expression values is a common practice during bulk RNA-seq analysis. This practice is meant to filter the data and eliminate background that could lead to statistical artifacts. We empirically set 10 as threshold value basing on both the distribution of our data and the background expression of negative control markers for our experimental setting.

Regarding the stringency of filtering parameters for RNA-seq analysis, we noticed an inconsistency of our data between the DEG list used for Figure 5c and 5d. We uniformed the results with a conservative strategy and fixed the main text accordingly.

6. The p and q values in both analyses in dataset 4 are highly repetitive within their respective columns. I'm not used to seeing this, so if it is normal and doesn't need fixing, please provide a statistical explanation for this.

This happens because the most recent versions of Cuffdiff (the software that we used for the differential expression analysis presented in this paper, see Methods section for further details) use a kind of permutation sampling procedure to assess the significance of differential expression. This is why the results present “bins" of significance values.

It would be useful if the authors could provide the complete results of the differential expression analysis for vsxKO compared to its respective control condition (wild type or one of the partial combinatorial mutants).

As mentioned in point 1, a complete list of DEGs for *vsx*KO compared to wild type has been added as a new worksheet in the Figure 5-source data 3 (former Supplementary Dataset 3).

In lines 465-474, the discussion highlighting the differences in Muller glia in the vsxKO compared to the mouse orJ mutant presents reasonable speculation as to the ultimate fate of the missing bipolar cells. A point for the authors to consider is that Rowan et al. (doi: 10.1016/j.ydbio.2004.03.039) noted that their Chx10 BAC reporter was primarily expressed in Muller glia when crossed into the orJ mutant. While they didn't quantify the proportion of Muller glia in the mutant, they did suggest that Vsx2 was preferentially expressed in Muller glia rather than rods.

We thank the reviewer for pointing to us the article by Rowan and Cepko, 2004 (PMID: 15223342); which further supports the proximity of the Müller glia and bipolar cells differentiation trajectory. We have now included this reference in the discussion: “Both glial and bipolars cells are late-born retinal types deriving from a common pool of precursors with restricted developmental potential (Bassett and Wallace, 2012; Hatakeyama et al. 2001; Rowan and Cepko, 2004; Satow et al., 2001).

In Figure 4 legend towards the end, the term "double mutant samples" is used. In keeping with the naming established earlier in the paper, it would be more consistent to refer to them as "vsxKO samples"

OK, done.